# A game-factors approach to cognitive benefits from video-game training: A meta-analysis

Evan T. Smith[1,2☯], Chandramallika Basak[1,2☯] *

1 Center for Vital Longevity, University of Texas at Dallas, Dallas, Texas, United States of America,
2 Department of Psychology, University of Texas at Dallas, Richardson, Texas, United States of America

☯ These authors contributed equally to this work.
* cbasak@utdallas.edu

## Abstract

This current study is a meta-analysis conducted on 63 studies on video-game based cognitive interventions (118 investigations, N = 2,079), which demonstrated a moderate and significant training effect on overall gains in cognition, $g = 0.25$, $p < .001$. Significant evidence of transfer was found to overall cognition, as well as to attention/perception and higher-order cognition constructs. Examination of specific gameplay features however showed selective and differential transfer to these outcome measures, whereas the genre labels of "action", "strategy", "casual", and "non-casual" were not similarly predictive of outcomes. We therefore recommend that future video-game interventions targeting cognitive enhancements should consider gameplay feature classification approach over existing genre classification, which may provide more fruitful training-related benefits to cognition.

## Introduction

Videogames have also become increasingly prevalent as a cognitive training medium, with controversial results regarding their effectiveness in facilitating transfer to a broad range of cognitive abilities [1, 2]. According to Simons et al. (2016), videogames implemented in cognitive training are typically divided into three broad categories: 1) "Action" games, which emphasize the rapid identification, prioritization, and handling of threats; 2) "Strategy" games, which emphasize resource monitoring and efficiently conducting multiple simultaneous tasks; and 3) "Causal" games, which are relatively simple and designed for short playtime compared to other categories. The distinction between these categories is an important one, as each theoretically encapsulates a separate construct of cognitive faculties [3–6], and therefore may produce different cognitive benefits after video game training (VGT). However, there is mounting evidence that such a coarse distinction between gaming genres is insufficient to describe the profile of cognitive demands of a given game, as modern video games increasingly include features of multiple genres [7]. Therefore, there is a need to examine VGT-induced benefits to various cognitive abilities based on the specific gameplay factors invoked by the games used.

Of these categories, "Action" games have been most extensively used in VGT [2], with reported cognitive benefits to visual attention [8–10], cognitive control [11, 12], and visual

**Funding:** This research was supported by a grant from the National Institute on aging to CB (titled

"Strategic Training to Optimize Neurocognitive Functions in Older Adults" under Award # R56AG060052; PI: Basak). The funder had no role in study design, data collection and analysis, decision to publish, or preparation of the manuscript.

**Competing interests:** The authors have declared that no competing interests exist.

working memory [13]. However, the "action game" moniker is incredibly broad regarding the types of games it has been applied to. Most commonly, studies of "action games" are referring to games in the "first-person shooter" (FPS) genre [14], with a handful referring to games in "racing" (i.e. *Need for Speed*), "fighting" (i.e. *Guilty Gear* [15]), "platformer" (i.e. *New Super Mario Bros.* [16]), and "sandbox" genres (i.e. *Grand Theft Auto V*, [17]). These genres differ widely from each other in terms of gameplay and mechanics. For example, both FPS and "racing" games involve first-person egocentric navigation of a three-dimensional environment, but differ in their primary method of interaction with the environment, namely combat (FPS) vs. movement (racing). Conversely, the "platformer" and "fighter" games cited above involve allocentric navigation through a two-dimensional, not three-dimensional, environment. They also differ from each other in terms of primary gameplay mechanic (platformers emphasize movement in avoidance of multiple hazards, whereas fighting games emphasize combat against a single opponent). The exemplar "sandbox" game uses egocentric navigation and combat interaction similar to FPS, but uniquely possesses multiple win-states that players are free to choose from, which is distinct from the single-goal structure of FPS, "racing", "platforming" or "fighting" games.

"Strategy" games are relatively understudied compared to "action" games [2], whose reported benefits include reasoning [18–20], visual short-term memory, working memory, and cognitive control [18]. Unlike the games classified under the "action" moniker, the games thus far classified as "strategy" games have almost universally belonged to one subgenre, "real-time strategy" (RTS) games. RTS games are played from an allocentric, third-person perspective similar to "platform" or "fighting" games (subtypes of "action" games), but are distinct from those actions games because the player is not bound to a single game object. Rather, RTS games require the player to simultaneously monitor multiple game objects to succeed. Some distinction can still be made between RTS games which feature an active antagonist (such as a rival civilization, as is the case with *Rise of Nations* [18]) from those that do not (as is the case with *Sushi-Go-Round* [3, 5]). Similarly, a distinction can be made between "strategy" games that feature multiple win-states (*Rise of Nations*) versus those featuring single goal (*Sushi-Go-Round* or *StarCraft*).

Research regarding the effectiveness of "casual" games is sparsest, with some evidence of benefits to attention and cognitive control [21, 22]. Complicating these results is the fact that the "casual game" designation is not an exclusive definition, but includes any relatively simple game which is designed to be played for short periods at a stretch (<30 minutes) and can potentially include many other genres of game. Indeed, Baniqued and colleagues (2013) examined the cognitive correlates of performance with 20 different casual games and found a wide range of correlation profiles. The granularity of cognitive constructs across casual games demonstrated by these studies justifies the experimental distinction between game genre—the correlation profile of "casual strategy" games differed qualitatively from that of "casual action" games—and also demonstrates that casual games should not be treated as a genre in their own right as the current literature tends to do [2].

If we assume that different gameplay factors reflect different cognitive demands, then these different games should be reclassified in terms of gameplay factors, not existing genres of "action" and "strategy"—yet the existing literature largely ignores this distinction. We hypothesize that the inconsistency regarding cognitive benefits from VGT is to some extent attributable to a lack of proper understanding of the differing cognitive demands evoked by gameplay factors. This oversight is particularly pronounced for "action" games. The aim of this meta-analysis is therefore to disambiguate the findings of the extant VGT literature by examining the specific gameplay features of the games used for training.

## Methods

### Literature search

A manual literature search was conducted for studies meeting the following inclusion criteria: video-game based cognitive intervention, pre- and post-intervention cognitive assessment, and randomized, controlled trials (RCT). Search terms included: "video game training", "[Genre] game training" (action, strategy, casual, real-time strategy, first-person shooter, racing, fighting, platforming), alternate spellings of these terms (i.e. "fighter", "shooter", "video-game", "computer game"), abbreviations (i.e. "FPS"), and substitution of "training" with equivalent terminology ("learning", "intervention", "cognitive intervention", "brain training"). S1 Method has the details on all search terms used. Studies with clinically impaired participants were excluded from this meta-analysis, given variability of the different clinically impaired populations and the limited number of video-game interventions therein. Studies of video-games specifically designed to facilitate cognitive transfer (i.e. "edutainment" games), as well as computerized interventions designed for cognitive remediation or maintenance, were excluded from this study, as the efficacy of such training methodologies has been examined in past work from the authors of the present meta-analysis [23]. Similarly, games designed as physical fitness activities (i.e. "exergames") were excluded from the present meta-analysis, due to the impossibility of disentangling the effects of gameplay and of physical activity on cognitive outcome of training. This literature search was conducted using the PubMed, MEDLINE, PsychARTICLES, PsychEXTRA, Psychology and Behavioral Science Collection, PsychINFO, PsychTests, and Google Scholar databases, with a cutoff date of May 1st, 2021. Reference lists of past meta-analyses and literature reviews [2, 23–31] were also used as a complement to broaden the literature search.

### Data collection

All included studies were read thoroughly and categorized based on the type of video-game intervention used, using the broad genre definitions provided in Simons et al. (2016) of "action", "strategy", and "casual" games. These were coded as two factors: Genre (*action* vs. *strategy*), and Format (*long-form* vs. *casual*), based on past research suggesting that the category "casual" video games encompasses games of multiple genres [3, 5, 6]. The Physiotherapy Evidence Database (PEDro) scale, an 11-item scale designed to assess quality and reporting of randomize controlled trials [32, 33], was used to assess quality of each individual study. The PEDro scale was utilized over other measures of study quality due to its extensive past use in high-quality meta-analyses of RCTs involving cognitive training [i.e. 23, 31, 34–36]. Variables were then coded for each study for subsequent moderator analyses. Four sets of moderators/variables were coded: 1) participant characteristics (% female, average age), 2) training characteristics (see below), 3) publication characteristics (year of publication, PEDro score, number of outcome measures), and 4) control group characteristics (see below).

Training characteristics included overall hours of video-game training, and a series of binary *gameplay factors* involved in the video-game. We included the following gameplay factors: Movement Style (*egocentric* vs. *allocentric* method of spatial navigation), Perspective (*1st person* vs *3rd person* viewing perspective), number of Controllable Objects (*single* vs *multiple*), number of Win States (*single* win state vs *multiple* win states), Type of Opponent featured by the game (*active opponent* vs *passive threshold)*, the presence of Time Pressure (*present* vs *absent*), and the Primary Interaction method of the game (*combat* vs *noncombat*).

The Movement Style, Perspective, and Controllable Objects were selected to typify the major gameplay features of the subgenres of "Action" and "Strategy" games most commonly

used in VGT interventions, namely "first-person shooter" and "real-time strategy" games [2, 14]. These two genres feature first-person perspective, egocentric movement, and a single controllable object (FPS), versus third-person perspective, allocentric movement, and multiple controllable objects (strategy). Time pressure was selected as a feature of interest due to the numerous past studies that have suggested that video games engender cognitive transfer via the application of high cognitive demands under time pressure [18, 24]. Combat versus an active opponent has been theorized to engender cognitive transfer by enforcing unpredictable task demands [1, 18–20, 24]—this was coded as the two factors of Combat and Active Opponent in order to flexibly capture games which featured combat versus an active opponent (i.e. the "FPS" and "RTS" genres), as well as games which featured unpredictable, active opponents but were not combat-focused (i.e. the "racing" genres). Lastly, Win States was selected as a feature of interest to test the assertion from Basak et al., 2008 [18] that the multiple win states possible in the *Rise of Nations* video game engendered transfer to task switching and executive function seen in that study, as well as to better categorize games such as *Grand Theft Auto V* [17] which allow for multiple play-styles and/or include many optional activities.

Movement Style was coded as *egocentric* if navigation in the game was performed with respect to the player's perspective (i.e. left/right movement), and *allocentric* if navigation was performed with respect to an outside reference (i.e. cardinal directions, the fixed edges of the play space). Perspective was coded as *1<sup>st</sup> person* if the information displayed to the player reflected the perspective of the player's game avatar, and was coded as *3<sup>rd</sup> person* if the player's avatar was visible from an outside perspective during the gameplay. Controllable Objects were coded as *single* if the player had simultaneous direct control of only a single game object/character, and classified as *multiple* if the player had simultaneous direct control of multiple game objects (such as the individual soldiers of an army). Win States were assessed by the number separate victory conditions that the player was able to pursue in the game. Games with a single victory condition (e.g. defeating an opponent, wining a race) were categorized as *single*, whereas games with multiple victory conditions (e.g. defeat *or* escape an opponent) were categorized as *multiple*. Type of Opponent was coded as an *active opponent* if success in the game depended on overcoming an unpredictable agent in the game environment, such as an AI-controlled enemy or obstacle. Static or purely predictable obstacles (e.g. the varied terrain of a Platformer game) were not considered *active opponents* for this purpose. Conversely, Goal Type was coded as *passive threshold* if success was determined by reaching or exceeding a static goal, such as a score threshold, completion time, distance traveled, etc., without the presence of an active opponent. Time Pressure was assessed as *present* if the player was required to make decision or take actions within a certain timeframe, else face negative consequences; these included games with time-sensitive action outcomes (such as scoring systems which take time into account) and games that proceed in real-time as the player acts. Time pressure was coded as *absent* if the player had an arbitrary amount of time to make decisions (i.e. turn-based games without turn time limits). Primary Interaction was coded as *combat* if attack or destruction of game entities or objects was the focus of gameplay, and *non-combat* if such content was not the focus of gameplay. As games were likely to include both combat and non-combat interactions, this factor was coded with respect to which interaction method occurred more frequently in the gameplay footage coded, as described below.

These gameplay factors were coded by four independent raters based on available gameplay footage of each game (see S1 Table for references to the analyzed footage). These raters were E. T.S. and C.B. Gameplay footage for two games (MultiTask and Smooth Snake) was not readily available—in these cases, raters coded gameplay factors based on direct experience with these games (see S1 Table). In cases where the categorization of a factor was ambiguous (i.e. games which feature some combat and some non-combat gameplay, games which were only partially

timed), raters were instructed to code that factor based on which feature was most prominent in the gameplay footage/gameplay experience analyzed. Inter-rater agreement of coded gameplay factors was 89.22%.

For each included study, the presence or absence of an active control group was coded. Since many studies included video-game interventions not only as experimental groups, but also as control groups, we further coded each video-game intervention as either experimental or control.

## Data analysis

The Effect Size of the Standard Mean Gain [37, 38] was calculated for every cognitive outcome in each intervention group, using the following formula:

$$ES_{sg} = \frac{\bar{X}_{T2} - \bar{X}_{T1}}{S_p} = \frac{\bar{G}}{S_g / \sqrt{2(1-r)}}$$

$$S_p = \sqrt{(S_{T1}^2 + S_{T2}^2)/2}$$

Where $\bar{X}_{T1}$ is the mean at time 1 (i.e., at pre-intervention baseline cognitive assessment), $\bar{X}_{T2}$ is the mean at time 2 (i.e., post-intervention cognitive assessment), $\bar{G}$ is the mean for time 2 ($\bar{X}_{T2}$) minus mean time 1 ($\bar{X}_{T1}$) gain score, $S_p$ is the pooled standard deviation of the gain score (calculated as above), $S_g$ is the standard deviation of the gain score, and $r$ is the correlation between the time 1 and time 2 scores. This is the method of effect size calculation recommended by Lipsey & Wilson (2001) [38] for pre-post contrasts without group comparison, originally sourced to Becker (1988) [37]. Because we did not have access to individual assessments for the interventions studied, $r$ was assumed to be 0.5. Directionality of each standardized mean gain score ($ES_{sg}$) was adjusted so that positive gain score indicates greater performance gain in the cognitive outcome from pre-intervention (Time 1) to post-intervention (Time 2).

Effect sizes ($ES_{sg}$) from all cognitive outcomes within each intervention were averaged into a single effect size estimate that represented the *Overall Cognitive Effect Size* from that specific intervention. This approach ensures maximum independency among the effect sizes [39], and that all studies are equally represented in every analysis despite the variation in the number of cognitive outcomes reported by each study.

In addition to the Overall Cognitive Effect Size, effect sizes were also calculated separately for four broad cognitive outcomes; viz. Attention/Perception (AP), Higher-order Cognition (HC), Memory (Mem), and Psychosocial (PS). The AP effect size aggregated measures of attention and/or perception, as well as pure reaction time measures—such measures have shown the most consistent sensitivity to video game training (particularly "action" game training) in past reviews of VGT interventions [2, 24]. The HC effect size aggregated measures of cognitive control, working memory, reasoning, and general executive function—all of which require effortful invocation of top-down control processes, and which are known to be highly inter-related [36]. These measures of higher-order cognition have been a strong focus of recent VGT interventions, but a conclusive answer as to the efficacy VGT in producing transfer to these measures remains contentious [2, 17, 26]. The Memory effect score aggregated measures of measures of recall and recognition for both short-term memory and long-term memory, and was of particular interest considering the possibility of VGT's efficacy as a cognitive remediation solution in older adults [23, 30, 31]. The PS effect size aggregated measures psychosocial wellbeing, personality and daily functioning. A list of measures included in each construct can be found in S2 Table.

After calculating these five effect sizes (*Overall*, *AP*, *Mem*, *HC*, *PS)* for each intervention, the standard error ($SE_{sg}$) and weight ($w_{sg}$) for each effect size from each intervention was calculated as follows:

$$SE_{sg} = \sqrt{\frac{2(1-r)}{n} + \frac{ES_{2n}^2}{2n}}$$

$$w_{sg} = \frac{1}{SE_{sg}^2} = \frac{2n}{4(1-r) + ES_{sg}^2}$$

where *n* is the common sample size at time 1 and time 2, and all other notations are as specified above. Overall $ES_{sg}$, $SE_{sg}$, and $w_{sg}$ for each included investigation are presented in S3 Table.

All meta-analyses presented below were conducted using mixed effects meta-regression models, with nuisance variables and regressors of interest modeled as fixed effects and varying by model as described below. The effect of study was modeled as a random effect in all analyses conducted below—this approach allows for random slopes and intercepts to be fit on a per-study basis, without treating investigations originating from the same study as independent. Heterogeneity in all analyses was estimated using a restricted maximum likelihood estimator method [40]. These analyses were facilitated by the *Metafor* package for the *R* coding environment [41].

## Results

### Characteristics of included studies

A total of 7,025 publications were reviewed during this literature search. 278 of these publications were identified as potentially of interest after abstract screening. Detailed screening for inclusion criteria reduced this number to 63 studies, representing 118 total interventions (Fig 1), with a combined *N* of 2079. A full list of these interventions can be found in S3 Table and S1 List.

After calculating the overall effect of each intervention, these studies were screened for outliers and removed on an individual intervention basis. Six interventions were removed for demonstrating an effect on overall cognition $</>$ 2 standard deviations from sample mean, resulting in a final *k* of 112 (*N* = 1917, see Fig 2) Of the remaining interventions, 90 reported Attention/Perception outcomes, 54 reported Higher-order Cognition outcomes, 28 reported Memory outcomes, and 18 reported Psychosocial outcomes.

### Relationship of genre to game factors, and correlations between game factors

The proportion of games of each coded Genre and Format (Action/Strategy, Long-Form/Casual) which featured each coded Game Factor is presented in Table 1. Notably, all of the games that our raters coded as *Serious* featured Time Pressure, and all of the games that our raters coded as *Casual* featured only a single win state.

Inter-correlations between the coded Game Factors are presented below in Table 2.

### Assessment of publication bias

Prior to performing our primary analyses, we first assessed publication bias in the sample of studies included in this meta-analysis. A regression test of funnel plot asymmetry [42] using the reported training gains to overall cognition as the variable of interest demonstrated significant asymmetry, $z = 6.41$, $p < .001$, indicating likely publication bias in this sample of studies.

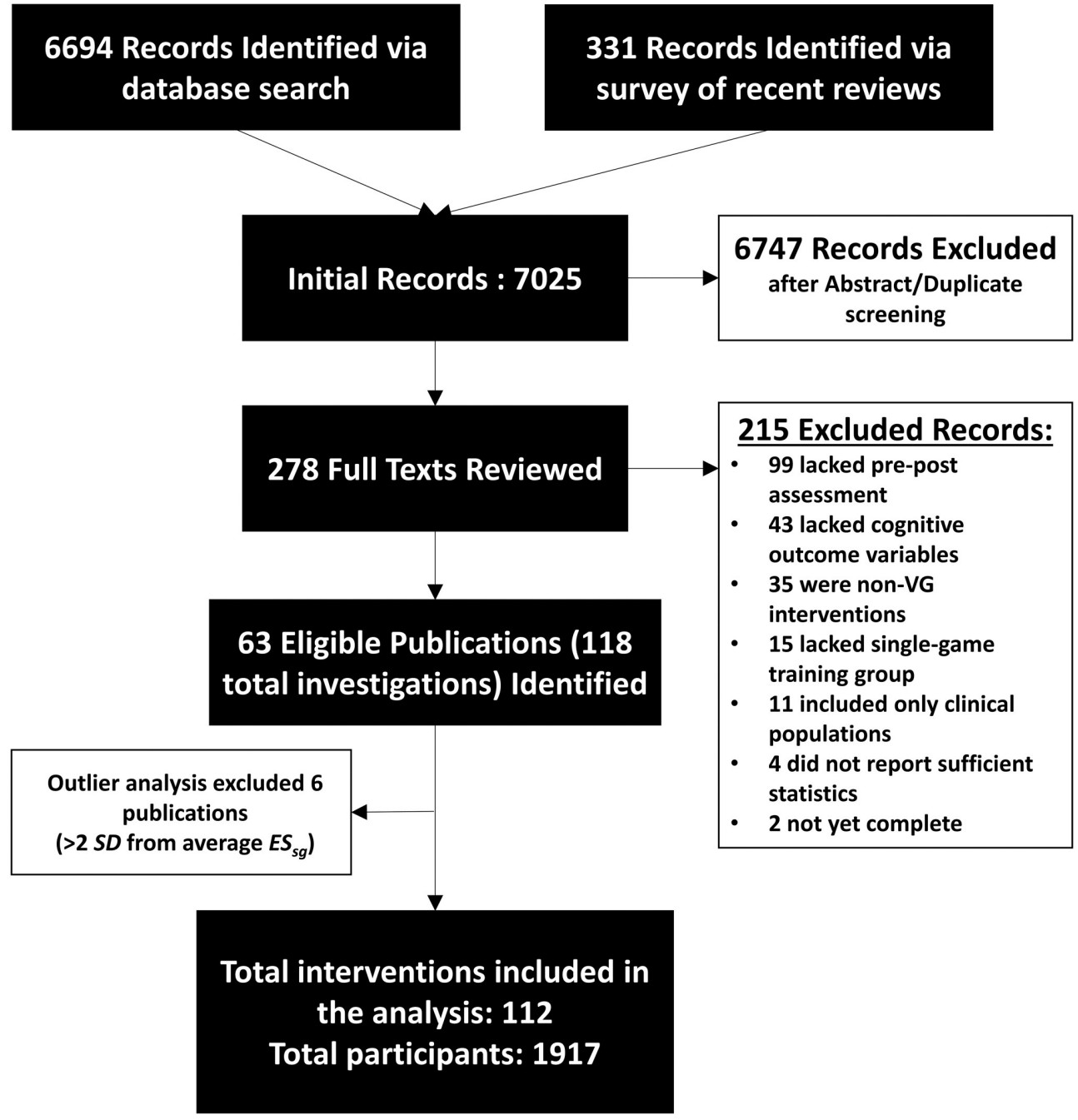

**Fig 1. Flow diagram of the systematic literature review conducted for this meta-analysis.**

A visual inspection of the funnel plot (see Fig 3) suggests that this bias is primarily in the positive direction, overestimating the effectiveness of VGT in relation to overall cognitive outcomes. To account for this $SE_{sg}^2$ was included in as a (fixed) nuisance term in all analyses. This method functionally extends Egger's regression test [42] to the multivariate modeling approach utilized in the present analysis, and thereby allows us to control for the detected asymmetrical influence of effect size on our outcome measure.

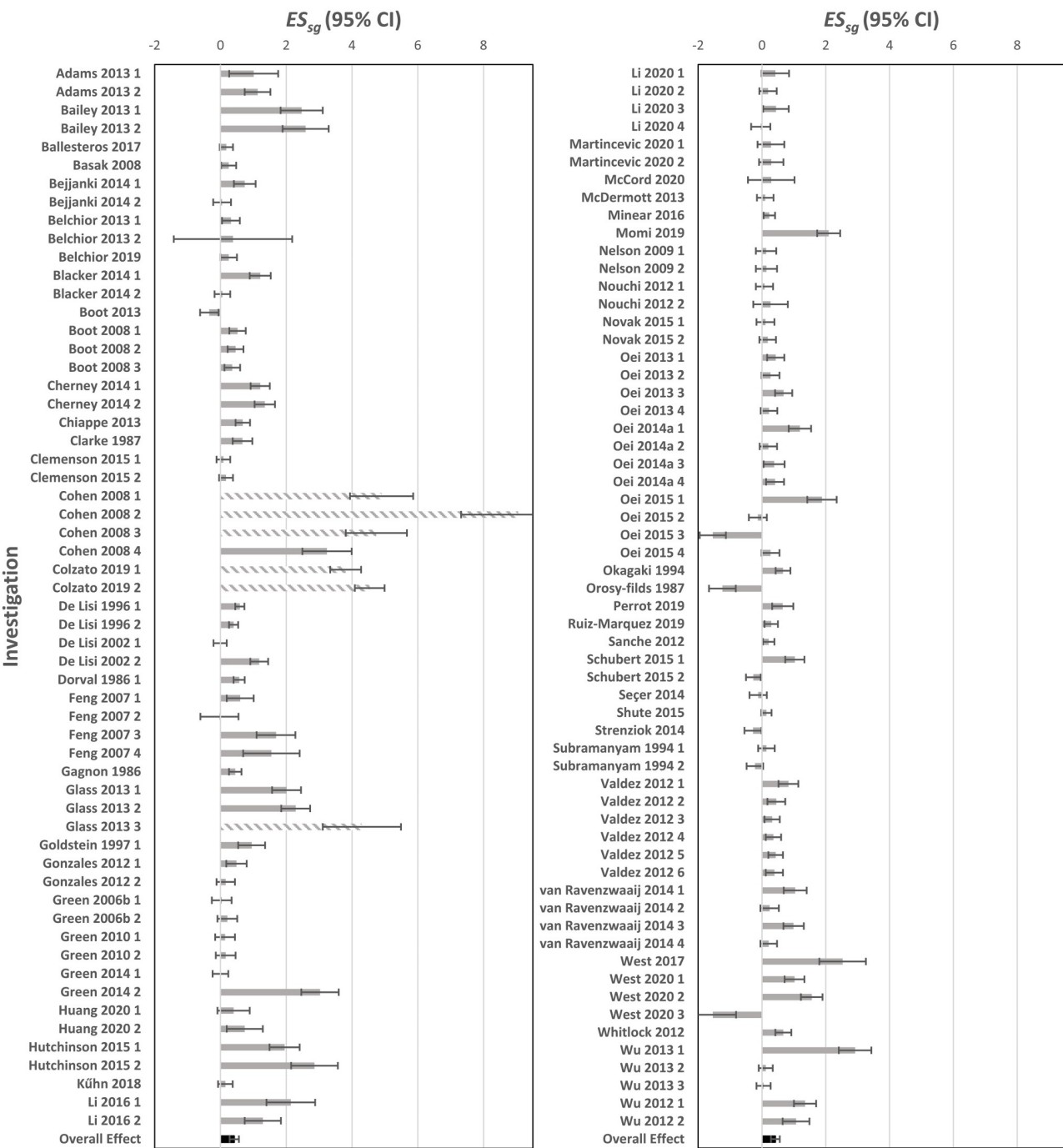

**Fig 2. Effect size of the standardized mean gain (ES_sg) on overall cognition for each included investigation.** *Note.* Partially shaded bars indicate studies that were excluded from analysis due to demonstrating an $ES_{sg} </>$ 2 standard deviations from sample mean.

## Overall effects of video-game training on overall cognition

The mixed meta-regression model predicting standardized mean gain to overall cognition (controlling for the fixed effect of investigation-wise standard error, as described above) Demonstrated a moderate and highly significant effect of VGT on overall cognition, $g = .25$, 95% CI

**Table 1. Percentage of games in each coded genre/format which featured each coded Game Factor.**

|  | Movement (Ego/Allo) | Perspective (1st/3rd) | Interaction (NC/Com) | Time (N/Y) | Objects (1/>1) | WinStates (1/>1) | Opponent (Pass/Act) |
|---|---|---|---|---|---|---|---|
| *Action* | 66/34% | 40/60% | 50/50% | 12/88% | 88/22% | 98/2% | 44/56% |
| *Strategy* | 13/87% | 7/93% | 60/40% | 20/80% | 27/73% | 67/33% | 60/40% |
| *Long-Form* | 71/29% | 41/59% | 36/64% | 0/100% | 79/21% | 86/14% | 24/76% |
| *Casual* | 22/78% | 17/83% | 83/17% | 40/60% | 65/35% | 100/0% | 91/9% |

[.12 .39], $p < .001$. Significant residual heterogeneity was detected in the sample, $Q(110) = 349.36$, $p < .001$.

## Effects of participant and study quality factors on cognition

To assess the impact on study quality on cognitive outcomes, we next fitted a mixed meta-regression model including the coded participant and study quality factors as fixed effect terms (with study modeled as a random effect as described above). Fixed factors included in this model included the gender ratio and average age of investigation participant (*% Female*, *Ave* Age), the PEDro score and total number of measures of each investigation (*PEDro Total*, *Num. Measures*), a binary variable coding for the presence of an active control group (Active Control), and a binary measure coding for weather or not the examined video-game intervention was considered as an active control group for another investigation in its study of origin (Study Group). Subject-wise sample variance ($SE_{sg}^2$) was included as a nuisance variable to control for the effects of publication bias. It was necessary to examine the Study Group factor considering we treated active control groups that utilized a video-game intervention method as groups of interest for the purpose of assessing the impact of their game factors for the present study. Comparing interventions used as training conditions versus interventions used as control conditions allowed us to test for experimenter bias for that were expected to produce a positive training outcome, and against games that were not.

The model examining the effects of participant and study quality factors on overall cognitive outcomes was found to be significantly predictive of overall cognition, $Q_w(8) = 50.72$, $p < .001$, $AIC = 267.71$. Study Group was the only variable of interest that significantly contributed to this model predicting overall cognitive outcomes, $\beta = -.32$, 95% CI [-.48, -.17], $p < .001$. As

**Table 2. Inter-correlations between coded gameplay factors, genre, and format.**

|  | Movement | Perspective | Interaction | Time | Objects | Win States | Opponent | Genre | Format |
|---|---|---|---|---|---|---|---|---|---|
| Movement |  | .57*** | -.33** | -.16 | .64*** | .13 | -.35** | -.46*** | -.48*** |
| Perspective | .57** |  | -.39** | .01 | .34** | .22 | -.26* | -.3* | .24 |
| Interaction | -.33** | -.39** |  | .29* | -.08 | .01 | .67*** | .08 | -.45*** |
| Time | -.16 | .01 | .29* |  | -.07 | .13 | .33** | .1 | .52*** |
| Objects | .64*** | .34** | -.08 | -.07 |  | .17 | -.13 | -.59*** | .15 |
| Win States | .13 | .22 | .01 | .13 | .17 |  | -.01 | -.46*** | -.24 |
| Opponent | -.35** | -.26* | .67*** | .33** | -0.13 | -0.01 |  | .14 | -.65*** |
| Genre | -.46*** | -.3* | .08 | .1 | -.59*** | -.46*** | .14 |  | -.05 |
| Format | -.48*** | .24 | -.45*** | -.52*** | .15 | -.24 | -.65*** | -.05 |  |

Note: single asterisk (*) indicates $p < .05$, double asterisks (**) indicates $p < .01$, and triple asterisks (***) indicates $p < .001$.

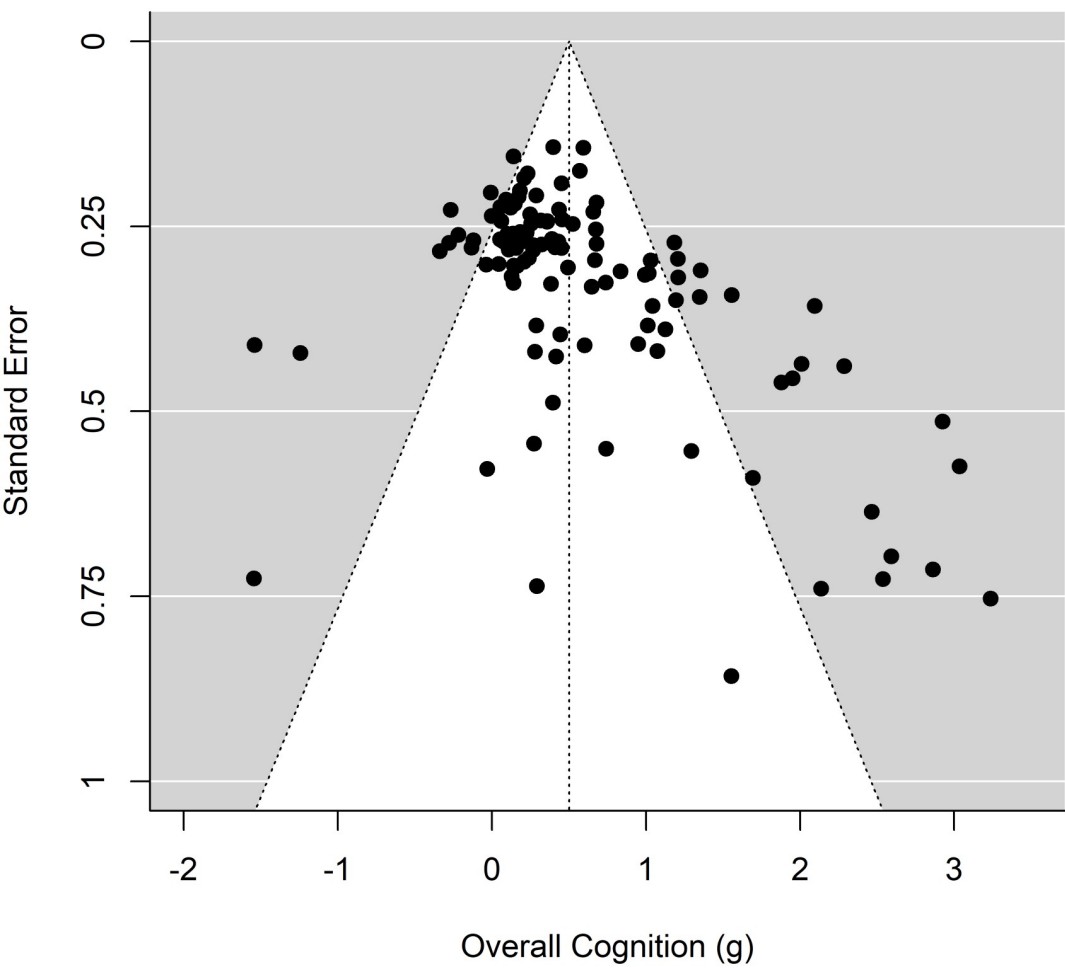

**Fig 3. Funnel plot plotting effect sizes of the standardized gains (Overall cognition), by Standard Errors for each included investigation.**

the Study Group variable was coded as *experimental group* = 0 and *control group* = 1, the negative valence of this result indicates that experimental group investigations produced significantly greater transition to general cognition than did control group investigations. The full fixed effects of this model are reported in Table 3.

### Effects of currently used genre distinctions on overall cognition

In order to assess whether the commonly used genre distinctions have differential impact on outcomes of video-game-based cognitive training, we next ran a mixed meta-regression model including those genre distinctions as fixed factors. Specifically, we included fixed factors for Genre [*strategy*, *action*] as well as for Format [*long-form*, *casual*]. As discussed above, we chose to separately categorize *action/strategy* and *long-form/casual* in this way due to past research which has demonstrated that games labeled as "Casual" in fact encompass a wide variety of gameplay styles and genres [3, 5, 6]. Additionally, in addition to the fixed effect of $SE^2_{sg}$, we also retained Study Group as a fixed control variable, considering it significantly contributed to the participant and study quality factors model.

**Table 3. Results of three LME meta-analysis models predicting overall cognitive outcomes of VGT interventions.**

| | $\beta$ | 95% ci lower | 95% ci upper | z | p |
|---|---|---|---|---|---|
| **Participant & Study Quality Model** | | | | | |
| [AIC = 258.7] | | | | | |
| $SE_{sg}^2$ | 2.13 | 1.36 | 2.89 | 5.44 | < **.001** |
| Ave. Age | -.01 | -.01 | .001 | -1.79 | .074 |
| % Female | -.0004 | -.01 | .005 | -.17 | .864 |
| Hours Total | .001 | -.01 | .01 | .29 | .768 |
| PEDro Total | .02 | -.12 | .16 | .28 | .783 |
| Num Measures | -.001 | -.02 | .02 | .04 | .965 |
| Active Control | .08 | -.23 | .39 | .49 | .627 |
| Study Group | .32 | -.48 | .16 | -3.87 | < **.001** |
| **Genre Model** | | | | | |
| [AIC = 258.87] | | | | | |
| $SE_{sg}^2$ | 2.17 | 1.43 | 2.92 | 5.71 | < **.001** |
| Study Group | -.29 | -.49 | -.1 | -2.89 | **.004** |
| Genre | .04 | -.22 | .13 | -.48 | .634 |
| Format | .06 | -.23 | .12 | -.65 | .519 |
| **Game Factors Model** | | | | | |
| [AIC = 252.87] | | | | | |
| $SE_{sg}^2$ | 2.04 | 1.26 | 2.81 | 5.22 | < **.001** |
| Study Group | -.24 | -.45 | -.02 | -2.1 | **.036** |
| Movement Style | .42 | .1 | .74 | 2.61 | **.009** |
| Perspective | -.47 | -.77 | -.18 | -3.12 | **.002** |
| Primary Interaction | .2 | -.05 | .44 | 1.57 | .117 |
| Time Pressure | .13 | -.14 | .4 | .94 | .349 |
| Controllable Objects | .06 | -.29 | .18 | -.49 | .627 |
| Win States | .02 | -.24 | .2 | -.19 | .849 |
| Type of Opponent | .09 | -.37 | .18 | -.67 | .5 |

**Note**: ci = confidence intervals; AIC = Akaike's Information Criterion

The model assessing the effect of Genre and Format on overall cognitive outcomes was found to be significantly predictive of overall cognition $Q_w(4) = 48.23$, $p < .001$, $AIC = 258.87$. However, neither the Genre, $\beta = -.04$, 95% CI [-.22, -.14], $p = .634$, nor Format, $\beta = -.06$, 95% CI [-.24, -.12], $p = .519$, significantly contributed to the model. These results suggest that these commonly used genre distinctions did not have differential effects on overall cognitive outcomes of VGT interventions. The full fixed effects of this model are reported in Table 3.

## Effect of gameplay factors on overall cognition

In order to assess if our coded gameplay factors produce separable effects on the overall cognitive outcome of VGT interventions, we next ran a mixed meta-regression model that included all of our coded gameplay factors as fixed effects (Movement Style, Perspective, Primary Interaction, Time Pressure, Controllable Objects, Win States, and Type of Opponents—see the Data Collection subsection of the Methods for a full description). As with the above model, fixed effects of $SE_{sg}^2$ and Study Group were included as control variables.

As before, this model examining the effects of the coded gameplay factors on overall cognitive outcomes was found to be significantly predictive of overall cognition $Q_w(9) = 60.6$, $p < .001$, $AIC = 252.87$. Both Movement Style, $\beta = .42$, 95% CI [.1, .74], $p = .009$, and Perspective, $\beta = -.47$, 95% CI [-.77, -.18], $p = .002$, contributed significantly to the model. The directionality of these effects indicates that games featuring *allocentric* movement were more efficacious in producing transfer to overall cognition that were games that featured *egocentric* movement, and similarly *first-person* games were more efficacious than producing transfer than were *third-person* games. The full fixed effects of this model are reported in Table 3.

## Effects of video-game training on cognitive and psychosocial outcomes

To test the possibility that different cognitive domains may benefit differently from video-game-based cognitive interventions, we next ran a series of meta-regression models examining the impact of VGT on specific cognitive outcomes of interest (Attention/Perception, Higher-order Cognition and Memory) as well as on the Psychosocial outcomes. As was the case with the models examining overall cognition, the models examining specific outcome constructs were controlled for the effects of investigation-wise standard error, and modeled random effects on a per-study (rather than per-investigation) basis. These models demonstrated significant training effects were demonstrated for Attention/Perception (AP) outcomes, $g = .27$, 95% CI [.08, .45], $p = .004$, and Higher-order Cognition (HC) outcomes, $g = .31$, 95% CI [.1, .51], $p = .003$. No significant training effects were observed for Memory outcomes, $g = -.14$, 95% CI [-.36, .06], $p = .17$, or Psychosocial outcomes, $g = .06$, 95% CI [.41, .53], $p = .812$. See Fig 4 for a visualization of these results.

## Effects of currently used genre distinctions on cognitive and psychosocial outcomes

Considering past research which has the "Action", "Strategy", "Casual", etc. game genres to different cognitive processes and profiles [2, 3, 5, 6, 13, 18], we next replicated analysis of game Genre and Format conducted above with respect to the *AP*, *HC*, *Mem*, and *PS* constructs. These analyses were conducted using an identical model to the Genre model used with respect to Overall Cognition, using each of these four constructs as a dependent variable in each respective analysis. This Genre model significantly predicted *AP*, $Q_w(4) = 50.16$, $p < .001$, $AIC = 220.4$, *Mem*, $Q_w(4) = 27.62$, $p < .001$, $AIC = 27.3$, and *PS* outcomes, $Q_w(4) = 18.6$, $p = .001$, $AIC = 20.3$, but neither the Genre nor Format term significantly contributed to any of those models (see the Genre Model section of Tables 2, 4 and 5). The *HC* construct was not significantly predicted by this model, $Q_w(4) = 4.01$, $p = .405$, $AIC = 128.51$.

## Effect of gameplay factors on cognitive and psychosocial outcomes

To assess the degree to which our coded gameplay factors are predictive of differential outcomes of cognitive training, and to compare the efficacy the game factors approach against the genre approach, we next replicated the Game Factors analyses reported above with respect to the *AP*, *HC*, *Mem*, and *PS* constructs. As above, the Game Factors model the model applied previously to overall cognitive outcomes applied to the *AP*, *HC*, *Mem*, and *PS* constructs in four separate analyses.

The Game Factors model significantly predicted the *AP* outcome construct, $Q_w(9) = 75.16$, $p < .001$, $AIC = 200.4$, with the slightly reduced AIC compared to the Genre model ($AIC = 220.4$) indicate a better model fit for the Game Factors model [43, 44]. The Perspective, Primary Interaction, Controllable Objects, and Type of Opponent terms all contributed significantly to the model (see Table 4). The directionality of these effects indicates that the *first*

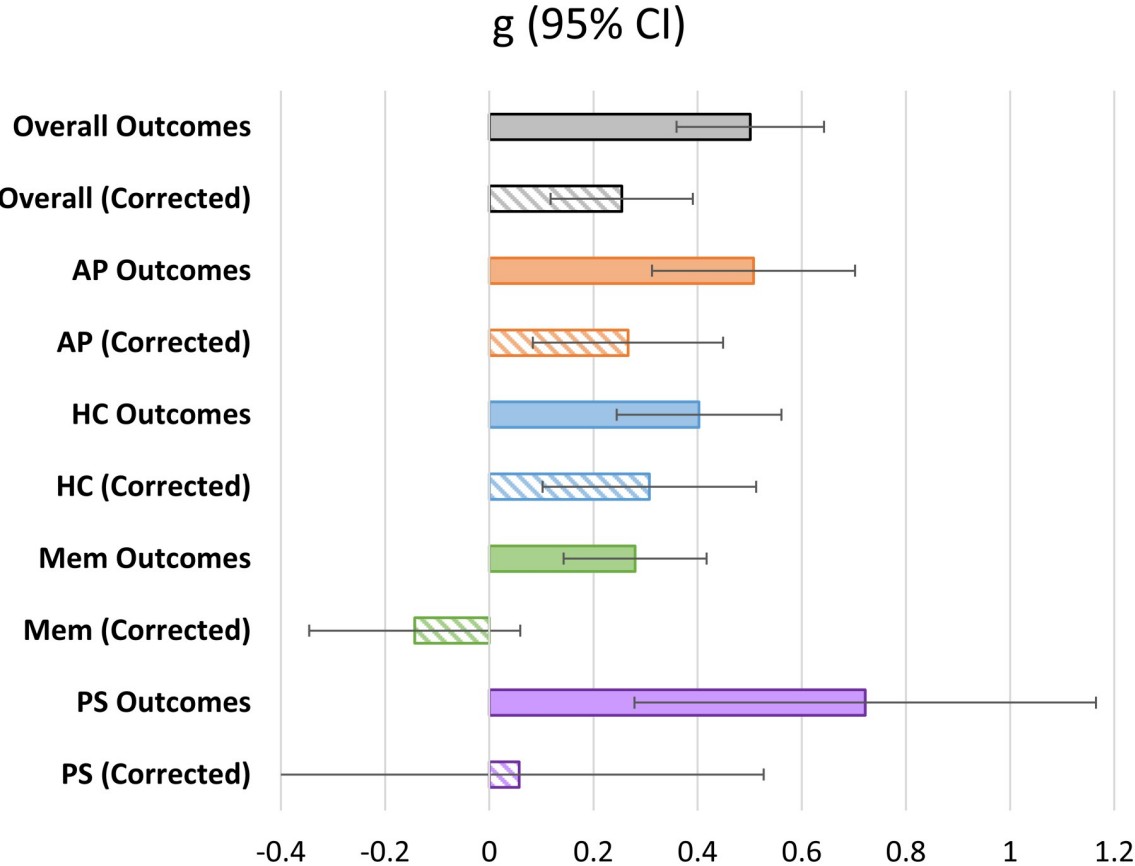

**Fig 4. Plot of effect sizes (g) & 95% confidence intervals for overall cognition and specific cognitive constructs (AP, HC, Mem, PS).** *Note*. Values labled as "Corrected" are corrected for publication bias via an extension of Egger's regression test [42] as defined in the Methods section above.

*person*, *combat*, *multiple controllable objects*, and *passive threshold* features were associated with greater transfer to attention and perception measures than the *third person*, *non*, *combat*, *single controllable object*, and *active opponent* features, respectively.

Similar to the above, the Game Factors model also significantly predicted the *HC* outcome construct, $Q_w(9) = 30.2$, $p < .001$, $AIC = 112.4$. Once again, AIC fell slightly from the Genre model ($AIC = 128.51$) to the Game Factors model, indicating a better model fit for the Game Factors model. Significant contributors to this model included the Movement Style, Primary Interaction, and Controllable Objects (see Table 5). The directionality of observed effects indicate that the $3^{rd}$-person, *combat*, and *single controllable object* features were related to greater HC outcomes than the $1^{st}$-person, *non-combat*, and *multiple controllable objects* features, respectively.

As with the *AP* and *HC* constructs, the Game Factors model also significantly predicted the *Mem* outcome construct, $Q_w(9) = 27$, $p = .001$, $AIC = 39.45$. However, this significant relationship appears to be driven entirely by the control terms included in the model—none of the gameplay factors significantly contributed to this model (see Table 6).

Lastly, the Game Factors model did not significantly predict the *PS* construct, $Q_w(9) = 14.24$, $p = .114$, $AIC = 25.94$, and again none of the included game factors were relevant to this model (see Table 7).

**Table 4. Results of three LME meta-analysis models predicting AP outcomes.**

| | β | 95% ci lower | 95% ci upper | z | p |
|---|---|---|---|---|---|
| | | **Genre Model**<br>[AIC = 220.4] | | | |
| $SE^2_{sg}$ | 2.24 | 1.34 | 3.14 | 4.89 | **< .001** |
| Study Group | -.4 | -.64 | -.16 | -3.25 | **.001** |
| Genre | .01 | -.21 | .23 | .13 | .9 |
| Format | -.04 | -.25 | .18 | -.34 | .731 |
| | | **Game Factors Model**<br>[AIC = 200.6] | | | |
| $SE^2_{sg}$ | 1.93 | 1.02 | 2.86 | 4.13 | **< .001** |
| Study Group | -.2 | -.46 | .06 | -1.54 | .124 |
| Movement Style | -.08 | -.56 | .39 | -.34 | .732 |
| Perspective | -.62 | -1.02 | -.21 | -2.95 | **.003** |
| Primary Interaction | .48 | .08 | .88 | 2.35 | **.019** |
| Time Pressure | .24 | -.15 | .62 | 1.2 | .23 |
| Controllable Objects | .4 | .03 | .76 | 2.1 | **.036** |
| Win States | -.05 | -.35 | .23 | -.41 | .682 |
| Type of Opponent | -.49 | -.89 | -.08 | -2.36 | **.019** |

**Note**: ci = confidence intervals; AIC = Akaike's Information Criterion

## Discussion

### Overall effectiveness of video-game training

The results of this meta-analysis indicate that, broadly, cognitive interventions utilizing video games are effective at producing training gains to general cognition, after taking publication

**Table 5. Results of three LME meta-analysis models predicting HC outcomes.**

| | β | 95% ci lower | 95% ci upper | z | p |
|---|---|---|---|---|---|
| | | **Genre Model**<br>[AIC = 128.51] | | | |
| $SE^2_{sg}$ | 1.13 | -.57 | 2.84 | 1.31 | .191 |
| Study Group | .05 | -.27 | .36 | .32 | .749 |
| Genre | -.03 | -.28 | .2 | -.32 | .748 |
| Format | -.19 | -.42 | .05 | -1.55 | .12 |
| | | **Game Factors Model**<br>[AIC = 112.47] | | | |
| $SE^2_{sg}$ | .77 | -1.04 | 2.59 | .83 | .404 |
| Study Group | -.04 | -.43 | .36 | -.17 | .682 |
| Movement Style | 1.37 | .81 | 1.94 | 4.79 | **< .001** |
| Perspective | -.3 | -.78 | .17 | -1.25 | .21 |
| Primary Interaction | .52 | .14 | .9 | 2.68 | **.007** |
| Time Pressure | -.03 | -.42 | .37 | -.13 | .896 |
| Controllable Objects | -1.01 | -1.45 | -.56 | -4.46 | **< .001** |
| Win States | -.1 | -.43 | .23 | -.56 | .574 |
| Type of Opponent | -.08 | -.48 | .31 | -.4 | .69 |

**Note**: ci = confidence intervals; AIC = Akaike's Information Criterion

**Table 6. Results of three LME meta-analysis models predicting memory outcomes.**

| | $\beta$ | 95% ci lower | 95% ci upper | $z$ | $p$ |
|---|---|---|---|---|---|
| **Genre Model** [AIC = 27.3] | | | | | |
| $SE^2_{sg}$ | 6.37 | 3.88 | 8.86 | 5.02 | < .001 |
| Study Group | .01 | -.34 | .34 | .03 | .977 |
| Genre | .04 | -.21 | .28 | .3 | .767 |
| Format | -.14 | -.44 | .15 | -.97 | .334 |
| **Game Factors Model** [AIC = 39.45] | | | | | |
| $SE^2_{sg}$ | 6.68 | 3.99 | 9.36 | 4.88 | < .001 |
| Study Group | -.16 | -.58 | .26 | -.75 | .451 |
| Movement Style | .18 | -.39 | .75 | .62 | .538 |
| Perspective | -.18 | -.68 | .31 | -.72 | .472 |
| Primary Interaction | -.01 | -.53 | .51 | -.05 | .963 |
| Time Pressure | -.18 | -.7 | .35 | -.66 | .511 |
| Controllable Objects | -.03 | -.52 | .46 | -.12 | .903 |
| Win States | .01 | -.41 | .44 | .07 | .947 |
| Type of Opponent | .09 | -.46 | .64 | .32 | .75 |

**Note**: ci = confidence intervals; AIC = Akaike's Information Criterion

bias into account. We must however emphasize that significant evidence of publication bias was found in this body of work, with published studies reviewed strongly favoring positive training outcomes even after accounting for anomalously large effects (see Fig 3). This is a known problem for interventions, including video-game trainings [2], which needs to be

**Table 7. Results of three LME meta-analysis models predicting PS outcomes.**

| | $\beta$ | 95% ci lower | 95% ci upper | $z$ | $p$ |
|---|---|---|---|---|---|
| **Genre Model** [AIC = 20.3] | | | | | |
| $SE^2_{sg}$ | 5.88 | .284 | 8.92 | 3.79 | < .001 |
| Study Group | .26 | -.65 | 1.17 | .56 | .576 |
| Genre | -.6 | -1.29 | .1 | -1.68 | .093 |
| Format | -.59 | -1.59 | .41 | -1.16 | .244 |
| **Game Factors Model** [AIC = 25.94] | | | | | |
| $SE^2_{sg}$ | 5.18 | 1.22 | 9.36 | 2.55 | .011 |
| Study Group | -.77 | -2.61 | 1.08 | -.81 | .416 |
| Movement Style | 1.51 | -1.85 | 4.87 | .87 | .379 |
| Perspective | -.23 | -1.64 | 1.08 | -.4 | .688 |
| Primary Interaction | .46 | -1.06 | 1.96 | .6 | .551 |
| Time Pressure | .11 | -1.96 | 2.18 | .1 | .918 |
| Controllable Objects | -.67 | -3.22 | 1.87 | -.52 | .604 |
| Win States | .2 | -1.3 | 1.69 | .26 | .797 |
| Type of Opponent | -.31 | -2.11 | 1.49 | -.34 | .735 |

**Note**: ci = confidence intervals; AIC = Akaike's Information Criterion

addressed in order to soberly assess the efficacy of such interventions as cognitive training. While the results presented here were corrected for publication bias, it is important that these findings be interpreted with the understanding that this evident publication bias may have magnified the effects observed.

There was also evidence of selection bias with regards to games used as training and/or the cognitive assessments selected as transfer methods. We treated all VG interventions as interventions of interest for our game factors analysis, even those designated as control conditions in their study of origin. Importantly, "control" condition games demonstrated less transfer to overall cognitive outcomes than did games examined as interventions of interest, despite a diverse range of games being presented as active control conditions (see S2 Table). Considering that the "control" games examined in the current meta-analysis include the vast majority of our coded gameplay factors (see Table 1), the diverse genre labels applied to these games in their studies of origin, and the varied cognitive demands of games even within the same genre [3, 7], a systematic difference in gameplay profile or cognitive demands between experimental and control games is an unlikely explanation of their difference in efficacy. The likely explanation here is one of design and task selection—the control games utilized in a given experiment by definition are those that are as similar to the experimental game as possible while retaining differences which the designers of the intervention expect will contribute to differential outcomes on their measures of interest. This finding, essentially, reflects successful selection of video games of differing gameplay profiles/cognitive demands to test specific experimental questions in the published literature. However, this approach does make disentangling the impact of specific gameplay factor—as we are trying to do in this meta-analysis—problematic, in that it is impossible to isolate single-factor differences between games that feature very distinct gameplay profiles, hence the necessity of the present meta-analysis considering all video game interventions regardless of possible status as a control condition as interventions of interest.

After considering (and correcting for where appropriate) these above caveats, the present meta-analysis still shows a significant and moderate effect on cognitive improvements from video-game training ($g = .25$; $p < .001$). While we cannot state with certainty that we have fully corrected for the observed publication bias, and we must acknowledge that these results are necessary uncontrolled due constraints in previous study design, we are confident that the observed effects are not spurious and that training with video games reliably produce transfer to cognition.

### Effect of specific gameplay factors on cognitive outcomes

Our primary interest in this meta-analysis was to examine the influence of specific gameplay factors on cognitive outcomes in video game training, since not all games are created equal. With regards to overall cognition, the present meta-analysis found that both perspective and movement style significantly influenced training outcomes. Specifically, training with games featuring a first-person perspective demonstrated greater cognitive transfer than those featuring a third-person perspective, and games featuring allocentric movement demonstrated greater cognitive transfer than those featuring egocentric movement.

While games utilizing these gameplay features are common in the field ("first-person-shooter" games utilize a first-person perspective, "real-time-strategy" and many "platformer" and "puzzle" games utilize allocentric movement), neither feature has been theorized to be particularly facilitative of cognitive training. Our findings here may be reflective of other gameplay features witch often co-occur with these two features. Specifically, the correlational analysis presented in Table 2 shows that the *Movement* and *Perspective* factors share a common pattern of correlation with the *Interaction*, *Objects*, and *Opponents* factors. It may very well be

that gameplay features coded for by these other factors are partially determinant of the finding that *Movement/Perspective* are impactful on overall cognitive outcomes. A further investigation of the complex relationship between these gameplay factors, including possible mediating and moderating effects, is warranted, and is a necessary step in further refining the game factors approach that we propose in this manuscript.

Interestingly, no single game examined in the present meta-analysis featured both an allocentric movement style and a 1[st]-person perspective, though in theory these results suggest that a game featuring may be a candidate for effective cognitive intervention, if we assume the effects of these two features are additive or synergistic. While they have not been studied within the context of cognitive interventions (insofar as the research reviewed within this meta-analysis), games that feature movement between pre-defined static first-person views such as *Five Nights at Freddy's* [45], *I'm on Observation Duty* [46], or *Observation* [47] include both gameplay features and would be ideal for testing this hypothesis.

**Attention & perception outcomes.** Past literature has reported transfer to attention and perception measures from "action" games as one of the more robust findings, which has been typically attributed to the strong emphasis placed on rapidly identifying and dispatching threats in "action" games [2, 5, 25, 48]. In terms of the game factors used in the present meta-analysis, the above definition of "action" games would translate to games that are combat-focused with time pressure, and have an active opponent. The current meta-analysis validated that combat-focused games were more effective in facilitating transfer to attention and perception than were non-combat focused games, partially validating past theorization. Additionally, games featuring a first-person perspective—one of the defining characteristics of the commonly-studied "first person shooter" subgenre of "action" games—demonstrated greater transfer than games featuring a third-person perspective. This finding agrees with the body of work which has found transfer to AP measures specifically resulting from such "first-person shooter" games [e.g. 6, 11, 49, 50]. However, games with passive thresholds or objectives better facilitated transfer to AP outcomes than did games with active opponents, and the AP construct was insensitive to the presence or absence of time pressure. These findings are somewhat surprising considering that "Action" games—defined in part by the need to rapidly identify the actions of unpredictable opponents—have consistently produced transfer to measures of attention and perception in past publications [2, 18, 25]. These findings suggest that neither active opponents or time pressure are necessary elements of gameplay to drive attention & perception outcomes—at least when considered in isolation.

Unexpectedly, games featuring multiple controllable objects produced greater transfer to attention & perception outcome measures than did games featuring only a single controllable object. Simultaneous control of multiple objects is a hallmark of the "real-time strategy" game subgenre of "strategy" games, and that feature has been theorized to drive transfer to working memory and cognitive control that has been observed in some intervention studies using such games [18]. The facilitation of transfer to attention and perception observed here is therefore unexpected, but may be attributable to the additional attentional demands of simultaneously monitoring several gameplay objects, compared to only a single gameplay object.

**Higher-order cognition outcomes.** The higher-order cognition construct used in this meta-analysis aggregated across measures of executive control, working memory updating, and reasoning. Transfer to this construct was found to be facilitated by games featuring allocentric movement (compared to egocentric movement), a primary combat interaction (compared to primarily non-combat games), and only a single controllable object (compared to multiple controllable objects). Only two of the games featured in interventions examined by the present meta-analysis featured all three of these game factors, those being the casual action games *Centipede* [51] and *Zaxxon* [52].

Importantly, some gameplay features which have been commonly theorized to drive transfer to higher-order cognitive functions such as reasoning and working memory were not indicated to be particularly efficacious by the results of this meta-analysis. As was the case with transfer to the AP construct, transfer to the HC construct was insensitive to the presence or absence of time pressure, which has been invoked as an important factor driving cognitive transfer from both "Action" and "Strategy" game interventions [18, 24]. From the body of work that has examined the "real-time-strategy" subgenre of "strategy" games, time pressure coupled with simultaneous control of multiple game object has been theorized to be an effective driver of transfer [18, 19]—though again our results do not support that conclusion, and indeed suggest that games featuring only a single controllable object are more effective in producing cognitive transfer. Conversely, combat games were more effective in producing transfer to the HC construct than were non-combat games, which is a core prediction of the action game training literature [2, 25]. It should be noted that the presence/absence of combat as a primary interaction, and presence/absence of time pressure were correlated to a significant degree in the sample of studies examined by this meta-analysis (see Table 2), so the significant influence of combat on transfer to HC measures may also account for the influence of time pressure in many combat-focused games.

**Memory & psychosocial outcomes.** Our coded game factors were not observed to impact transfer to memory or psychosocial outcome measures, and indeed video-game training was not found to reliably transfer to either outcome construct after accounting for publication bias. However, it is also important to note that only 25% (28) and 16% (18) of studies examined by this meta-analysis examined memory and psychosocial outcomes, respectively. While we did not find evidence of transfer to either of these constructs in the present meta-analysis, these results could be the result of a relative paucity of studies which report memory or psychosocial outcomes coupled with highly varied intervention methodologies of the cognitive intervention literature [2], and/or the relatively coarse outcome constructs examined in the present meta-analysis.

### Relevance of existing genre definitions and recommendations for future research

The "action" versus "strategy" categorization utilized in this meta-analysis proved ineffective in distinguishing cognitive outcomes of VGT-based cognitive interventions. As can be seen in Table 1, these two genres featured a heterogeneous mix of gameplay features, reminiscent of the wide array of gameplay styles featured in the "action" game literature specifically and as discussed above in our literature review, so it is perhaps unsurprising that the distinction between these two genres was not reflected in differential cognitive outcomes. However, it should be noted that the binary "action" versus "strategy" categorization utilized in this study is an exaggeration of how those terms have been classically applied. Yes, both the label of "action" (and to a lesser extend "strategy") have been applied to a disparate array of games, past studies have not purported that these two genres could be used to broadly classify *all* games, with varying other labels being applied to different sections of the gameplay space (i.e. "puzzle" or "simulation") [2, 53–56]. Further, those studies that do focus on the effect of "action" or "strategy" games often utilize specific subgenre terms which are more suggestive of the overall cognitive demands of gameplay (i.e. "first-person shooter" or "real time strategy", rather than "action" or "strategy" [2, 14, 19, 57]. It is conceivable that a more specific categorization of game genre would result in more consistent training outcomes and/or training outcomes more reflective of genre-standard gameplay features. However, considering that a) the division between game genres has become increasingly irrelevant as that field of art has

progressed to hybrid genre and other novel formats [7], and b) games within the same genre can have very different cognitive demands [3, 4], even more specific genre classifications have limited utility. We therefor recommend that a more thorough description of the gameplay features present in a training game be presented in future research, in lieu of classification by genre.

This proposed approach presents several benefits. Firstly, an understanding of specific gameplay features can aid the field in interpreting mixed findings that VGT studies have produced thus far. Secondly, specification of gameplay factors can aid in the design of future studies, by isolating specific gameplay features that improve intended outcomes and by creating more rigorous control groups that are matched with the training group on potentially confounding gameplay features. The gameplay classification system presented in this analysis is more rigorous than any previously applied in the field, but also represents the first attempt to create such a system. In particular, the approach taken in this manuscript is limited by a-priori definition of game factors and the lack of consideration of conjunctive effects of these features. Further development thoroughly-researched classification systems to describe the specific gameplay demands of VGT interventions—and particularly how those gameplay demands relate to cognitive demands—would serve as a boon to future video-game-related research.

The "Long-Form" vs "Casual" game distinction likewise appears to have little relevance in terms of relation to observed training outcomes. "Casual" games do not appear to be a distinct genre featuring a distinct combination of gameplay features, but rather a format of video-game that can be applied to games of multiple genres [3]. The current meta-analysis did not find meaningful difference in cognitive transfer between studies utilizing "Long-Form" vs "Casual" games, suggesting this may not be a useful distinction with regards to cognitive impact of training paradigms using these genres. On the other hand, this lack of distinct effects between "Long-Form" and "Casual" games may prove a useful tool in the design of future video-game based interventions. Specifically, assuming a given investigation is not concerned with long-term memory or psychosocial outcomes, shorter-duration Causal games can be utilized to reduce the consecutive time required for training sessions, allowing for more design flexibility in terms of participant and experiment scheduling, as past research from our group has leveraged [5, 6]. Note that the authors make no claim regarding *overall* training duration necessary to produce cognitive transfer as we found no significant relation between training duration and cognitive outcome in the analyses conducted—our above statement specifically pertains to the utility of "Casual" games for use in individual training sessions.

## Registration

This meta-analytic review's protocol and analysis plan was pre-registered at https://osf.io/apmk5. We followed the PRISMA-P checklist when preparing the protocol, and we followed PRISMA reporting guidelines for the final report (see S1 File for PRISMA checklist). The meta-analytic data are shared at https://osf.io/6j792/.

## Supporting information

**S1 Method. Full search terms for literature search.**
(DOCX)

**S1 Table. Gameplay factors of games featured in included studies.**
(DOCX)

**S2 Table. Outcome measures categorized by construct, with representative studies.**
(DOCX)

**S3 Table. Characteristics of included video-game training studies.**
(DOCX)

**S1 List. References for included studies.**
(DOCX)

**S1 File. PRISMA checklist.**
(DOCX)

## Author Contributions

**Conceptualization:** Evan T. Smith, Chandramallika Basak.

**Data curation:** Evan T. Smith.

**Formal analysis:** Evan T. Smith, Chandramallika Basak.

**Funding acquisition:** Chandramallika Basak.

**Investigation:** Evan T. Smith, Chandramallika Basak.

**Methodology:** Evan T. Smith, Chandramallika Basak.

**Project administration:** Evan T. Smith, Chandramallika Basak.

**Software:** Chandramallika Basak.

**Supervision:** Chandramallika Basak.

**Validation:** Chandramallika Basak.

**Visualization:** Evan T. Smith, Chandramallika Basak.

**Writing – original draft:** Evan T. Smith.

**Writing – review & editing:** Chandramallika Basak.

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
