## [Decision Letter · Decision Letter 0]

31 May 2022

PONE-D-22-06647A Game-Factors Approach to Cognitive Benefits from Video-Game training: A Meta-AnalysisPLOS ONE

Dear Dr. Basak,

Thank you for submitting your manuscript to PLOS ONE. I am sorry for the relative delay in the evaluation of your paper. I had initially found two reviewers to assess the paper, but one reviewer fell through. I then had to search for a new reviewer. Gladly, I found one more expert that agreed to assess the paper, and I am glad I did because this expert provided excellent comments on how to improve your manuscript. I would like to take this opportunity to thank both reviewers for doing a great job in assessing the paper and providing many constructive comments. As you will see from the comments appended below (and include as attachment to this email), the reviewers have made a number of comments regarding your proposed classification and your analysis method, both in terms of needed clarifications but also with regards to improvements of the analytical methods used. I strongly recommend you to carefully consider each of their comments and helpful suggestions. I will not reiterate all of their points here. I will only make a few remarks that arouse on my own reading of the paper (see below). Overall, I believe that there is a clear revision path that may lead to publication, so I am inviting a major revision. Note that this invitation does not guarantee acceptance of the paper and hence you should do your best effort to address all concerns of the reviewers (as well as the points I will present below). My own comments:First, Similarly to Reviewer 2, I thought that the evaluation of the paper was compromised by some accessibility issues regarding the preregistration and the data. With regards to the preregistration, the project is private and hence we could not read the preregistration. The text also does not indicate which hypotheses were preregistered neither which analyses are part of the preregistration and which ones were exploratory. The text should be revised to clearly identify these. With regards to the data, it seemed to be some sort of image of an excel file, which unfortunately is not useful for sharing data since one cannot reuse the data (unless one retypes all of the information visually available). I wanted to explore the data-set, but one can neither download it, see it closely, click on it or anything. Please be aware that sharing of data should follow the FAIR principles (findable, accessible, interoperable, and reusable). These two points will be certainly critical in subsequent evaluation of your paper, if you decide to submit a revision.

Second, it stroke me that most the of the factors included in the analyses of the games that were not significant, still generated effects that went in the same direction as the significant ones, often with similar mean size (but sometimes even larger) although lacking significance. Given that these cases were also often the cases with less studies (small k), it seems to me that the difference between significant and non-significant is probably not very informative. This should be taken in consideration. Can one be certain that one factor is more conducive than the other in generating a transfer effect? Please consider this issue carefully. One possible approach to mitigate this issue would be to perform a Bayesian Meta-analysis and derive uncertainty estimates. Alternatively, you should provide the reader with some sort of evidence of which differences observed with the current analysis are meaningful, and provide the caveats explaining the ones that are yet not credible (for example, due to the lack of sufficient studies included). Thank you for considering PLOS ONE as the outlet for your work.

We look forward to receiving your revised manuscript.

Kind regards,

Alessandra S. Souza, Ph.D.

Academic Editor

PLOS ONE

Journal Requirements:

3. We note that you have stated that you will provide repository information for your data at acceptance. Should your manuscript be accepted for publication, we will hold it until you provide the relevant accession numbers or DOIs necessary to access your data. If you wish to make changes to your Data Availability statement, please describe these changes in your cover letter and we will update your Data Availability statement to reflect the information you provide

Reviewers' comments:

Reviewer's Responses to Questions

**Comments to the Author**

1. Is the manuscript technically sound, and do the data support the conclusions?

Reviewer #1: Partly

Reviewer #2: Yes

2. Has the statistical analysis been performed appropriately and rigorously? 

Reviewer #1: Yes

Reviewer #2: I Don't Know

3. Have the authors made all data underlying the findings in their manuscript fully available?

Reviewer #1: Yes

Reviewer #2: Yes

4. Is the manuscript presented in an intelligible fashion and written in standard English?

Reviewer #1: Yes

Reviewer #2: Yes

5. Review Comments to the Author

Reviewer #1: Review of A Game-Factors Approach to Cognitive Benefits from Video-Game training: A Meta

Analysis by Smith and Basak

Overall evaluation: Meta-analyses are tedious work and usually always criticized for the many decisions the authors take during working on inclusion and exclusion criteria. I appreciate the work invested by the authors and as usual have some remarks regarding decisions taken by the authors which are, in my reading not enough motivated.

1, why is the physiotherapy evidence database chosen and were studies excluded based on this assessment?

2, why are age and percentage female chosen as moderators as well as publication characteristics – I think that overall moderator choice deserve a better motivation

3, to the gameplay factors: again, I miss a somewhat motivated choice why those factors ( Movement Style (egocentric vs. allocentric method of spatial navigation), Perspective (1st person vs 3rd person viewing perspective), number of Controllable Objects (single vs multiple), number of Win States (single win state vs multiple win states), Type of Opponent featured by the game (active opponent vs passive threshold), the presence of Time Pressure (present vs absent), and the Primary Interaction method of the game (combat vs noncombat) were chosen, ideally those choice can be tied to cognitive processes that are assumed to be affected by the engagement in playing those games. Finally, I wondered whether movement style and perspective code different factors. Thus, did the authors correlate their game factors or run a cluster/factor analysis to uncover shared and unique variance contributed by those factors.

4, the authors did not motivate their choice for the outcome measures either or do give any examples (what is an attention/perception outcome and why is it not a HC outcome).

As stated in the beginning, meta-analyses are tedious work and I presume the authors really invested lots of time and work but for the time being, to many choices remain unmotivated or opaque. Overall, I see a lot of merit in the consideration of gameplay factors instead of game category.

Reviewer #2: The study provides an interesting meta-analytic approach to study the impact of video game training on cognition, focusing on game features that are thought to characterize distinct video game genres. This is an original approach that will make an interesting contribution to the field and hopefully should stimulate further research. I have some clarification questions and also some suggestions that aim at strengthening some results and conclusions.

The present meta-analysis was pre-registered on OSF and I had access to the PRISMA checklist but not to the full methods. Therefore, the comments below are based on the information provided in the manuscript only, without taking the pre-registered methods analysis plan and hypotheses into consideration.

My primary concern relates to the use of within-subject effect sizes, which do not properly control for numerous confounds such as practice or test-retest effects or placebo effects. While I understood that this was necessary to include control games, the use of effect sizes based on pre-post change scores has important limitations that should be acknowledged (Cuijpers et al., 2017). My second and third main concerns are related to the analytic strategy (assuming independent effect sizes, using r = 0.5) and the lack of details regarding the criteria used for coding game features, genres, formats, and cognitive constructs. I felt that the paper would benefit from clarifying both the choice of categories (genre vs format, experimental vs control) and the levels (action vs strategy, serious vs. casual).

However, given that the study was pre-registered, I wasn’t able to determine if there were deviations from the registered methods (coding), analysis plan (effect sizes and models) and hypotheses tested (number of analyses). If an analysis appears sub-optimal but was registered, any change or any additional analysis, even if it represents an improvement, should be strongly justified and yet considered exploratory as it will deviate from the registered analysis plan.

Below I provide more details about each specific point. I am confident that the authors can address most of them in a reply, and that providing some of the additional analyses suggested in this review will strengthen their results and thereby increase the impact of their study. (see attached document)

6. PLOS authors have the option to publish the peer review history of their article (what does this mean?). If published, this will include your full peer review and any attached files.

Reviewer #1: No

Reviewer #2: No

---

## [Author Response · Author response to Decision Letter 0]

11 Oct 2022

Response to Editor and the Reviewers

We sincerely thank the reviewers and the Editor for their insightful comments and we have addressed all of these comments in the revised manuscript that has undergone substantial changes. 

Please find our specific responses below, where all comments are in Italics and our responses are in regular font:

Editor Comments

1. First, Similarly to Reviewer 2, I thought that the evaluation of the paper was compromised by some accessibility issues regarding the preregistration and the data. With regards to the preregistration, the project is private and hence we could not read the preregistration. The text also does not indicate which hypotheses were preregistered neither which analyses are part of the preregistration and which ones were exploratory. The text should be revised to clearly identify these. With regards to the data, it seemed to be some sort of image of an excel file, which unfortunately is not useful for sharing data since one cannot reuse the data (unless one retypes all of the information visually available). I wanted to explore the data-set, but one can neither download it, see it closely, click on it or anything. Please be aware that sharing of data should follow the FAIR principles (findable, accessible, interoperable, and reusable). These two points will be certainly critical in subsequent evaluation of your paper, if you decide to submit a revision.

a. The accessibility issues expressed by the editors and reviewers were not intended by the authors – we are fully committed to providing access to the dataset utilized in this study. The difficulties in access the shared dataset appear to stem from accessibility restrictions enforced by the authors’ home university, which we believe we have addressed. The shared materials should now be fully viewable and editable by any party with an access link – please contact the corresponding author (Chandramallika Basak) if this is not the case, as further access issues are erroneous.

2. Second, it stroke me that most the of the factors included in the analyses of the games that were not significant, still generated effects that went in the same direction as the significant ones, often with similar mean size (but sometimes even larger) although lacking significance. Given that these cases were also often the cases with less studies (small k), it seems to me that the difference between significant and non-significant is probably not very informative. This should be taken in consideration. Can one be certain that one factor is more conducive than the other in generating a transfer effect? Please consider this issue carefully. One possible approach to mitigate this issue would be to perform a Bayesian Meta-analysis and derive uncertainty estimates. Alternatively, you should provide the reader with some sort of evidence of which differences observed with the current analysis are meaningful, and provide the caveats explaining the ones that are yet not credible (for example, due to the lack of sufficient studies included).

a. After carefully considering this comment (as well as those from reviewer 2 who expressed similar misgivings re. our statistical approach), we decided to re-analyze our data using a mixed-effect meta-regression approach – see lines 253-257 of the revised method section, as well as the revised publication bias analysis (lines 301-310). This approach allows us to a) compare and correct for all of our variable of interest in a single model, b) reduces the total number of comparisons (a consequence of point a), c) directly compare the effect of subgroups on our outcome measures, and d) incorporate a correction for publication bias (via Egger’s regression test) into the models natively. We feel this revised approach addresses all of the major concerns provided by the reviewers and the editor, and is a much stronger statistical approach overall. 

Reviewer 1 Comments:

1. why is the physiotherapy evidence database chosen and were studies excluded based on this assessment?

a. More information regarding the PEDro scale as well as justification for its use have been added in lines 134-138

b. This was not an exclusion criteria, but was examined as a factor of study quality.

2. why are age and percentage female chosen as moderators as well as publication characteristics – I think that overall moderator choice deserve a better motivation

a. they are not – average age and percentage female are participant characteristics, distinct from publication characteristics. However, this distinction is no longer relevant in the revised statistical approach taken in the re-submitted manuscript – participant characteristics, publication quality characteristics, and all of the binary factors considered in the initial analysis (genre, format, study group, game factors) have been modeled as fixed effects in a mixed-effect meta-regression model (see lines 253-60 and lines 324-338)

3. to the gameplay factors: again, I miss a somewhat motivated choice why those factors ( Movement Style (egocentric vs. allocentric method of spatial navigation), Perspective (1st person vs 3rd person viewing perspective), number of Controllable Objects (single vs multiple), number of Win States (single win state vs multiple win states), Type of Opponent featured by the game (active opponent vs passive threshold), the presence of Time Pressure (present vs absent), and the Primary Interaction method of the game (combat vs noncombat) were chosen, ideally those choice can be tied to cognitive processes that are assumed to be affected by the engagement in playing those games. Finally, I wondered whether movement style and perspective code different factors. Thus, did the authors correlate their game factors or run a cluster/factor analysis to uncover shared and unique variance contributed by those factors.

a. Justification for why each gameplay factor was chosen has been added to the methods section (line 148-164), and a specific call to refine the game factors approach with respect to specific cognitive demands has been added to the discussion (lines 614-617). We agree with reviewer one that establishing how specific gameplay factors influence specific cognitive demands would be a major boon to the cognitive science of video games and VGT, though establishing that experimentally would first require a standardized approach to categorizing gameplay which doesn’t exist within the field at the moment – and which this meta-analysis is intended to facilitate.

b. Re. movement style and perspective: Perspective codes weather the player perspective is from within an actual or implied player avatar (1st-person) or if that avatar is visible to the player (3rd-person). These factors are correlated (see the additional correlation analyses added to the results section, lines 287-290 & Table 2) but are not redundant. 

i. For more concrete examples, the present analysis included 1st-person games with egocentric movement (i.e. Call of Duty, Unreal Tournament, Portal), 3rd-person games with egocentric movement (i.e. Grand Theft Auto V, Mario Kart, World of Warcraft) and 3rd-person games with allocentric movement (i.e. Pac-man, StarCraft, The Sims). It does not appear that 1st-person games with allocentric movement have yet been studied within the field, though such games do exist (see the examples given on lines 512-513, all of which feature first-person perspectives coupled with allocentric map-based navigation, i.e. the player can chose their first-person vantage point by selecting that point from a list or an allocentrically-presented map).

4. the authors did not motivate their choice for the outcome measures either or do give any examples (what is an attention/perception outcome and why is it not a HC outcome).

a. Further justification for the construction of our outcome constructs have been added to lines 229-244

¬¬¬¬-___________________________________________________________________________________

Reviewer 2 Comments:

1. Confounds. The first relates to the interpretation and presence of confounds. As is now standard in intervention studies, the use of a control group is essential to rule out potential confounds, such as placebo, test-retest or practice effects. This is also true in meta-analyses that aim to compare different groups or interventions.

a. The reviewer is correct that utilizing a control group is essential for controlling for many sources of noise in a study, including in meta-analyses. Ideally, we would have calculated a relative gain measure for all examined interventions versus an active control condition. However, considering that the following:

i. A substantial portion of the examined publications utilized video games as an active control group (i.e., Belchior et al., 2013; Blacker et al., 2014; Clemenson & Stark, 2015; Cohen, Green, & Bavelier, 2008; De Lisi & Wolfard, 2002; Feng, Spence, & Pratt, 2007, etc.), or examined differences in transfer across multiple video games without specifying one specifically as a control condition (i.e., Adams, 2013; Bailey & West, 2013; Boot et al., 2008; Glass, Maddox, & Love; 2007; Gonzales, 2012; Huang, 2020; Oei & Patterson, 2013; 2014a; 2015; etc.)

ii. Video games utilized as a control condition must have a different gameplay profile to serve as a suitable active control to a video game intervention of interest 

iii. Creating a differential effect score comparing relative gain on two different games with two different profiles of gameplay features necessarily confounds the distinct gameplay features of the two games compared

b. Considering the above, it is not possible the primary experimental question of this review – that is examining the differential effects of gameplay factors on cognitive training – using differential measures which necessarily confound different gameplay features. This is a limitation, but a necessary one to address the issue at hand.

c. The mixed meta-regression approach presented in this re-submitted manuscript (lines 253-260, Tables 3-7) does not account for common confounds with regards to our estimation of the overall effect of game training, but it does allow us to definitively compare the effectiveness of subcategories defined by the coded game factors (i.e. combat vs noncombat games, etc.) which partially addresses this concern.

2. Between group comparisons. Second, the use of within-subject effects also makes between-group comparisons less reliable as the Q statistics controls for only one source of heterogeneity related to within-group variance and not for the variance of between-group differences in pre-test performance, post-test performance or change scores, that are essential for comparing subgroups. 

a. The mixed meta-regression approach presented in this re-submitted manuscript (lines 249-252, Tables 3-7) allows for direct subgroup comparisons. The discussion has been re-written to reflect the findings of these direct comparisons.

3. Pre-post correlation. A third problem arises from the correlation between pre-test and post-test which is almost never reported in primary studies and thus calls for adjustments of the effect sizes and their variance. While assuming a prepost correlation of r = 0.5 is a neutral (i.e., unbiased) choice, this choice will impact the estimates, their variance (hence the weights) and thereby the results. It would be reassuring to see sensitivity analyses testing the impact of different values of r and this would certainly strengthen the results and conclusions. This is particularly important given that the meta-analysis is based on within-subject change scores and the correlation between pretest and postest can thus be expected to be higher than 0.5. Yet, sensitivity analyses testing several values below and above 0.5 would seem advisable.

a. While a sensitivity analysis would allow for the assessment of how different estimates of r would affect the results, it wouldn’t allow for a more accurate estimation of the “correct” estimate of r for pre-post correlations in this sample. Yes, we could assume that pre-post scores are highly correlated due to the within-subjects nature of the comparison, but that assumption is impossible to confirm with the data available. We thought it prudent to select a single unbiased value for r (=.5, which the reviewer noted as well) and run the analysis only once, rather than select between r values based on patterns of results.

4. Random effects model. The authors use random effects models, which assume dependent effect sizes. Yet, the shared excel files (which are hardly readable and would be much more useful as downloadable excel files) suggest that the authors extracted more than one effect size from each study and group of participants. Were effect sizes coming from the same study aggregated or randomly selected and how? If averaged, how was variance handled? If randomly selected, did the author assess sensitivity via bootstrapping? Because effect sizes from the same study may not be fully independent (even if coming from different groups), an alternative may be to use methods that can handle dependent effect sizes, such as multilevel methods or meta-analytic models using robust variance estimates which allow to specify a correlated or hierarchical structure of the weights.

a. The “unreadability” of the shared excel file possibly suggests that the reviewer was viewing a preview of the file, not the file itself. We recommend downloading the file (upper-left corner on most browsers) which will grant the reviewer access to the excel file in full.

i. Difficulty accessing the file (i.e. if there is no “download button) may be the result of security measures put in place by our parent university. We will happily supply this file directly (via the Editor to maintain anonymity) if the issue persists, but we believe we have set access permissions sufficiently to avoid this problem.

b. Multiple interventions/groups from the same study were treated as separate studies in the initial analysis and a separate effect size was calculated for each, as specified on lines 224-228 of the manuscript.

c. In line with this reviewer’s suggestion, we have taken a mixed effect meta-regression approach in the resubmitted manuscript, with the effect of study of origin modeled as a random effect in all analyses. Modeling the random effect by study rather than by intervention/group ensures that the inherent inter-relation of investigations arising from the same study is accounted for in the revised modeling.

5. Moderators were assessed across multiple subgroup analyses. Were all analyses pre-registered and if yes did the authors consider correcting for multiple testing? One drawback of this approach is that it assesses the impact of each moderator individually, without controlling for other moderators, some of which may covary (e.g., genre, format, features, participants age). Meta-regression models can assess the influence of several moderators simultaneously. Meta-regression was used to assess continuous moderators and a similar approach may be recommended at least for some analyses of categorical (or dummy coded) moderators. Finally, some subgroup analyses may be underpowered as they rely on relatively small numbers of studies or effects.

a. As stated above, we have we have taken a mixed effect meta-regression approach in the resubmitted manuscript, in line with this reviewer’s suggestion. As suggested by this reviewer, these mixed meta-regression models include both the study and participant quality moderator as well as categorical moderators, allowing for the effects of each to be controlled for the others, and reducing the total number of comparisons substantially.

6. Publication bias analyses are limited to trim and fill, which has been extensively criticized. There are a multitude of methods available now, each with its own pros and cons, that are generally recommended over the trim and fill approach. An increasingly common approach is to run publication bias sensitivity analyses using a variety of techniques (Carter et al., 2019; Mathur & VanderWeele, 2020). Moregenerall, I highly recommend these two recently published meta-analyses which provide state-of-the-art methods in terms of meta-analytic models for dependent effect sizes, moderator analysis and publication bias detection (Coles et al., 2019; Lehtonen et al., 2018).

a. We removed the trim-and-fill method as this reviewer suggested, and instead corrected for publication bias using a generalization of Egger’s regression test as explained on lines 307-310. This approach was chosen specifically because it could be integrated with the mixed effect meta-regression models used in the re-submitted manuscript.

b. We agree that correcting for publication bias with regards to p-values rather than study variances, as suggested by Carte et al., 2019 and Mathur & VanderWeele 2020 is likely a more ecologically valid method of correcting for publication bias, and an approach that should be considered seriously by the field. However, as far as we are aware such methods have not been implemented in statistical packages in such a way that also allows for multi-level or mixed-effect meta-regression models (the approach taken in the re-submitted manuscript, also recommended by this reviewer in questions R2.4 and R2.5). Existing statistical packages (i.e. the “PublicationBias”, “puniform” packages for R) only allow for this method to be applied to simpler regression models. There is clearly a need for development of such a statistical package, but we unfortunately lack the coding expertise to develop such an approach in the timeframe of this review process.

7. The introduction and discussion largely argue that the existing literature has emphasized an association between video game features and their cognitive effects and the present study attempts to address this hypothesis. Yet, I believe that the coding could be improved to better reflect the authors’ hypotheses as coding was sometimes inconsistent with the authors argument. Some games could be arguably coded in a different genre and some tasks in different constructs which might change the results and conclusions. 

a. The coding of games into broad genres is problematic – even in the implementation we used in this meta-analysis – and this is a major impetus for conducting this meta-analysis, as discussed in the other responses to Reviewer 2.

b. The justification for the coding of cognitive constructs is explained in our response to question R2.13

8. Coding of game features. The part I liked most concerns the coding of game features which I found particularly interesting and could have been pushed even further. For example, a theory-grounded coding mapping specific game mechanics with particular cognitive demands (rather than mechanics associated with genres) would probably allow a different set of testable predictions. 

a. We agree with reviewer 2 that the coding of game features can and should be further developed – one of the impetuses for this meta-analysis is to spur such future work. The authors assert that the field has yet to develop a thorough enough understanding of which gameplay features are associated with what cognitive capacities to allow for such a comprehensive categorization strategy, and that developing such a strategy as game cognition research continues to mature is of strong importance. 

9. Coding of game genre and format. What criteria were used to define genres and format? casual and serious games have been considered as video game genres in other studies, suggesting these 2 categories may overlap at least partially. Aren’t action, strategy, casual and serious meant to be mutually exclusive categories? How was the game Tetris (from Belchior et al) classified? It appears in both action and casual — which would seem wrong to me? How about Portal 2? Was it considered an action game and on which criteria? Were serious games only commercial games (how about cognitive training like cogmed, lumosity)? How about brain training games? Are these considered casual or serious? These notions should be clarified. Note that some codes (e.g. format, serious) were missing from the shared word doc and that the excel files were hardly readable or usable (no possibility to search). Also, the word “commercial games” is mentioned for the first time in the discussion line 453. Does that refer to off-the-shelf games? Was that an inclusion criterion? Were all serious games commercial games? There have been debates and inconsistencies in how genre was defined and handled in prior meta-analyses (either as selection/inclusion criteria or as moderator) and I am not clear if and how format differs from genre. This is a central point of the manuscripts that deserved some clarification including whether and why these codes are considered independent. 

a. Our response to this question has been organized into multple parts to address the multiple issues that Reviewer 2 raised in this comment

i. “What criteria were used to define genres and format?” 

1. The distinction between “Genre” (action|strategy) and “Format” (“serious|casual”) was based largely on work by Banequed and colleagues (Banequed et al., 2013; 2014; Smith et al., 2020), which demonstrates that “casual” video games fall into the same genre distinctions (i.e. “action, strategy”) as their non-casual counterparts. We therefore categorized both serious and casual games as either action or strategy. This is specified on lines 130-132 of the Methods section.

ii. “casual and serious games have been considered as video game genres in other studies, suggesting these 2 categories may overlap at least partially. Aren’t action, strategy, casual and serious meant to be mutually exclusive categories?”

1. “Action” and “Strategy” are mutually exclusive in our coding strategy, as are “Serious” and “Casual”. However, all games are categorized on both of these variables (so a game may be “Action” and “Serious”, “Action” and “Casual”, and so on).

2. “Serious” in the context of this meta-analysis simply refers to a game that is not a “casual” game (as defined on line 131 of the methods section) and therefore these are mutually exclusive designations

iii. “How was the game Tetris (from Belchior et al) classified? It appears in both action and casual — which would seem wrong to me?”

1. Both Tetris and Super Tetris were categorized as a “Casual" and “Strategy” games by our raters. According to supplementary table 1

2. As discussed above, all games were categorized as either “Action” or “Strategy” and either “Casual” or “Serious” - this dual categorization is not an error.

iv. “How about Portal 2? Was it considered an action game and on which criteria?”

1. Both Portal and Portal 2 were categorized as “Serious” and “Strategy” by our raters. Again, this dual categorization is intended

v. “Note that some codes (e.g. format, serious) were missing from the shared word doc and that the excel files were hardly readable or usable (no possibility to search).”

1. The “Serious” vs “Casual” distinction (I.e. Format) was original summarized by the “Casual Game” column of Supplementary Table 1. This has been re-labeled as the “Format” column, with each game listed as “C” (Casual) or “S” (Serious) to address this confusion

2. The inability to zoom or search on the shared excel file was an unintentional effect of the security protocols used in UTD’s file sharing service. We have taken steps to remedy this. If difficulty accessing this file continues, we would happily provide a copy directly (via the Editor, to preserve anonymity)

vi. “Also, the word “commercial games” is mentioned for the first time in the discussion line 453. Does that refer to off-the-shelf games? Was that an inclusion criterion? Were all serious games commercial games?”

1. The “commercial games” wording was an artifact of a previous draft of the discussion section which was erroneously retained in a submitted draft. Third has been corrected

vii. “Were serious games only commercial games (how about cognitive training like cogmed, lumosity)? How about brain training games? Are these considered casual or serious? These notions should be clarified.”

1. We did not examine cognitive training or brain training games in this meta-analysis, as we feel that cognitive impact of those games is qualitatively different than games meant as entertainment and therefore not directly comparable. This has been specified on lines 117-120.

viii. “There have been debates and inconsistencies in how genre was defined and handled in prior meta-analyses (either as selection/inclusion criteria or as moderator) and I am not clear if and how format differs from genre. This is a central point of the manuscripts that deserved some clarification including whether and why these codes are considered independent.”

1. The authors strongly agree with Reviewer 2’s statement here, and hope that the responses included in this document and changes to the manuscript have clarified this issue.

10. Also, the genres Action and Strategy may be (arguably) too restrictive to capture all games. Forcing all games to fit into these 2 genres results in categories that are too heterogeneous to be meaningful. For example, the Action genre may not be the best fit for games like Centipede, Donkey Kong, Pacman or Wii Fit… Similarly, games like Angry Birds, The Sims or Tetris do not seem to fit well into the Strategy genre. Others have considered categories like simulation, exergame or platform games or even brain training. The literature search mentions brain training games but I didn’t see any. Were brain training games excluded and why? And if not how were these games coded? 

a. Indeed, the Action/Strategy distinction is inadequate to categorize all games (the authors opinions on the topic broadly align with Dale & Green, 2017, as indicated in lines 48-53 of the introduction).

b. The inclusion of the Action/Strategy coding scheme alongside the more detailed Game Factors coding approach was designed contrast (one of) the ways Genre has been classified by the field in the past with our newer approach. We do not endorse the Action/Strategy distinction, and as Reviewer 2 notes we argue against the utility of that classification strategy at several points. Thus, the authors observe no contradiction in its use in this study as a point of comparison to the more detailed classification method we propose here.

c. The distinction between simulation, strategy, platforming, and action games is poorly defined throughout the literature, as discussed extensively in our introduction (lines 54-84). Our argument is that any such genre distinctions are by-and-large arbitrary and unhelpful – therefore the two-genre action/strategy categorization approach is just as valid (or invalid) as a 4-genre approach which also includes puzzle or platformer games.

d. Related to the above, we instructed our reviewers to categorize games as “action” or “strategy” as they deemed fit – this was an enforced binary choice. 

e. Exergames and brain training games were excluded from the present analysis. This has been specified on lines 116-122

f. Related to the specific example cited by Reviewer 2: according to supplementary table 2, Angry Birds, Centepede, Donkey Kong, Pac Man, and Pac Man: Adventures in Time , were coded as “action” games by our raters, while The Sims, The Sims 2, The Sims 3, Tetris, and Super Tetris were all coded as “strategy” games by our raters. Wii Fit, being an exergame, was excluded from the present analysis in line with the above comment.

11. More importantly, the categorization of games into genres seems inconsistent with the authors' argument that games that belong to a given genre share features that are responsible for certain cognitive effects. 

a. See response b to comment R2.10

12. Coding of experimental and control games. Here too, the reader would benefit from a more detailed description of how this coding was performed as it wasn’t available in the supplementary excel file. Note that the coding of serious games is also missing… Again, these are important as the criteria for defining serious games or experimental vs control games has varied across studies.

a. Games were not universally coded as “experimental” vs “control” - this was a binary factor of the intervention examining a game, not of the game itself, as specified on lines 206-208 of the manuscript. If the condition from which an effect size was generated was an experimental condition in its study of origin, that case was labeled as an “experimental” study (see supplementary table 2). Similarly, if the condition from which an effect size was generated was a control condition in its study of origin, that case was labeled as an “experimental” study (again, see supplementary table 2)

13. Coding of cognitive constructs. Relatedly, a similar reflection would also be relevant for defining cognitive constructs of interest. What criteria guided the choice of these 3 outcome categories: attention/perception, higher cognition and psychosocial? Previous meta-analyses of video games (and in other fields too) have categorized cognitive measures into separable constructs may provide some guidance or inspiration (e.g. Bediou et al., 2018; Powers et al., 2013; Powers & Brooks, 2014; Sala et al., 2018; Wang et al., 2017). What is the rationale for including some memory (working memory) tasks in higher cognition, whereas others are coded as memory? Psychosocial also includes depression and anxiety scales, affect or emotion (PANAS), risk taking… which may not really correspond to psychosocial skills. Again, the categories appear too heterogeneous to be meaningful. For example, memory may include tasks that include manipulation of verbal or visuospatial material which may be differentially impacted by distinct games. Studies of memory may also include older participants and although the effect is NS, the trend is negative suggesting smaller effects in older participants (consistent with previous meta-analyses that treated age as categorical that report smaller effects in older adults).

a. Further justification for the construction of our outcome constructs have been added to lines 229-244. Working memory was specifically coded as Higher Cognition and not memory because of its known inter-relation with reasoning and executive function (see Miyake et al., 2020)

14. The authors conclude that game features predict differential effects on cognition. Yet, their effect sizes do not differentiate between skills as the confidence intervals overlap and the use of within-subject effects does not allow proper between-group comparisons. Therefore, some conclusions may need to be toned down or rephrased and these limitations should be better acknowledged. 

a. As stated above, our revisited statistical approach does allow for direct subgroup comparisons. The discussion section has been re-written with respect to only those differences found to be significant with the revised approach to analysis.

15. Mapping between features and cognitive constructs. The game features coded do not clearly map onto specific cognitive demands, which appears to contradict what the authors discuss about the choice of experimental and control groups in primary studies. Lines 465-477, “Importantly, “control” condition games failed to produce any significant transfer to any cognitive construct, despite a diverse range of games being presented as active control conditions (see Supplementary Table 1). This is evidence that, in general, VG-based interventions have tailored their cognitive transfer measures specifically to be sensitive to the cognitive demands of the intervention, and/or have intentionally selected control games not expected to produce transfer to the constructs examined. While this is an understandable design strategy, this practice may lead authors to incorrectly conclude that the control games used, or aspects thereof, do not produce cognitive improvements. As a concrete example of why this is problematic, consider the games Tetris and The Sims. Both games have been frequently used as active control conditions in studies of “action” video games, and producing negligible transfer as expected [12, 13, 37]. However, both of those games have demonstrated significant transfer, in studies in which they were an intervention of interest.”

a. This statement makes no claims as to the link between cognitive constructs and specific game features – the lack of transfer to ANY cognitive outcomes from Control games (which, as established by this meta-analysis, have a multitude of gameplay profiles spanning the coded gameplay features) preclude gameplay features as an explanatory factor of this lack of transfer. We suggest that a form of selection bias is present, in which experimenters are selecting control games which they believe a-priory will not cause transfer to their outcome measures.

b. This statement has been re-worded in light of this comment and the findings from our adjusted statistical approach (see lines 473-488)

16. The cases of Tetris and The Sims are particularly illustrative and interesting as the choice of using these as experimental or control games may depend on which aspect of cognition was measured (and considered as demonstrating transfer). This seems to contradict another statement that appears a few lines later. Lines 500-510: “the above definition of “action” games would translate to games that are combat-focused with time pressure, and have an active opponent. However, the current meta-analysis found consistent transfer to measures of attention and perception was facilitated by games that featured a first-person perspective, a single controllable object, a single win state, lacked an active opponent, and were primarily non-combat games. Not only was transfer to the attention and perception construct found to be agnostic to the presence of time pressure, but both active opponents and combat gameplay explicitly failed to produce consistent transfer. This finding provides evidence that the post-hoc justifications given in the past explaining the link between “action” video game play and enhanced attention and perception may be inaccurate.”

a. The referenced statement has been substantially altered in light of the findings from our adjusted statistical approach as well as this comment (see lines 519-534)

17. These are strong statements and conclusions which depend largely on the appropriate coding of game features, video game genres and cognitive construct, as well as their appropriate analysis allowing more direct comparisons which were not possible here. 

a. As stated above, our statistical approach has been substantially altered and now does allow for direct subgroup comparison – the resubmitted discussion is written with regards to the findings of this updated statistical approach

18. Finally, the sentence line 566-567: “The gameplay factors analysis conducted in this meta-analysis rebuffs the “action” vs “strategy” distinction as useful.” appears too strong and should be downplayed given the results and the rest of the discussion arguing that this distinction has some value. This study sheds some light on how some of the game features that are present in action and/or strategy games and may be associated with their (differential?) cognitive effects. However, I don’t think it rebuffs the distinction between action and strategy games… As the authors rightfully discuss on line 573, the “genre distinction does have merit” (as well as some “historical” value) as action and strategy games were initially quite distinct in terms of mechanics, features and cognitive demands, and thus represented meaningful categories less than 10 years ago. However, as the video game ecosystem grows and new genres, hybrid genres and mixed genres arise, the features that were once specific to action or strategy games have become more and more common across genres and thus more difficult to isolate (see (Dale et al., 2020; Dale & Green, 2017). Toning down the conclusion won’t minimize the potential impact of this study which will make an important contribution to the field and should stimulate more work looking at how game characteristics map onto cognitive demands and whether they predict cognitive effects. 

a. The conclusion re the utility of genre distinction has been re-worded in light of this comment to be less definitive and more accurately reflect the limitations of this work (see lines 580-585). We should note that the evidence supporting the partial merit of genre distinction was in fact weakened by our adjusted statistical approach, which did impact our interpretation of these results.

19. The authors argue that the lack of effect in the control condition “is evidence that, in general, VG-based interventions have tailored their cognitive transfer measures specifically to be sensitive to the cognitive demands of the intervention.” (line 468). This conclusion would require an analysis based on the cognitive demands rather than the game characteristics –which were not coded here– unless there is a perfect match between game features (e.g. pace or time pressure) and cognitive demands (attentional control)?

a. Our argument re the possible influence of selection bias on the relative efficacy of experimental vs control condition games has been re-worked, see lines 470-484

20. Study quality / risk of bias. Details about the PEDro scale would be appreciated, especially because it is less commonly used than other tools such as the Cochrane risk of bias scale. (https://guides.himmelfarb.gwu.edu/systematic_review/reporting-quality-risk-of-bias). 

a. More information regarding the PEDro scale as well as justification for its use have been added in lines 134-138

21. Participant and study characteristics. Meta-regressions show non-significant modulation by either participant or study characteristics. Were these moderating influences also considered in subgroup analyses? Some factors tend to covary (e.g., sample size, age, training duration) and may thus be difficult to disentangle.

a. The participant and study characteristics were not considered in the subgroup analyses presented in the original submission, but are included in the mixed effect meta-regressions in the revised manuscript.

22. Participants characteristics: Are groups matched in gender? Proportion (%) females overall can be misleading if the groups are not balanced. Why was average age treated as continuous… previous work has mostly used age-group categories (e.g., children, young adults, old adults).

a. The majority of studies examined either reported no difference in gender between experimental groups or explicitly gender matched the groups (see table 2). Some had disproportionately more males across all groups (i.e. Adams 2013, Di Lisi & Cammarano 1996), and some explicitly compared the effects of training between genders (i.e. Feng, Spence & Pratt, 2007; Subrahmanyam & Greenfield, 1994). This disparity between studies, and in some cases between groups in the same study, is why we included % female as a moderator of interest.

b. Average age was treated as continuous for several reasons. Firstly, age bin cut points are not consistent across the field (i.e. some studies may categorize participants 60 and older as “older adults”, some 65 and older, etc.). Second, even if age bit cut points are consistent, there are considerable differences in cognitive profile and social cohort between individuals in the same age bin – i.e. an 80-year-old “older adult” would be expected to have a substantially different cognitive profile than a 60-year-old “older adult”, while a 20-year-old “younger adult” would be of a substantially different social cohort compared to a 40-year-old “older adult” especially with regard to video games and technology. Using average age as a continuous variable allows us to preserve variance that may correspond to these differences.

23. Study characteristics: Number of participants is already included in the weights (inverse variance) and therefore I don’t think it needs to be added to the models. What was the rationale for coding the number of cognitive outcomes? What result was expected? It is surprising that total hours had no effect but again there may be complex interactions with participant characteristics, cognitive construct, game features or genres.

a. Re number of participants: this point is well taken - n was not included in the meta-regression models due to the noted redundancy with the weights used.

b. Re number of cognitive outcomes: Since a single effect size of each construct was calculated from each study regardless of the number of outcome measures, controlling for that number accounts for systematic differences in effect size that this variance of measures may induce, such as relative susceptibility of each effect size to the task impurity problem (see Miyake et al., 2000). Previous work from our group has found such systematic differences (Basak, Qin & O’Connell, 2020).

c. Re total hours of training: The submitted manuscript first meta-analysis to show negligible gains from longer training durations – see Basak, Qin, & O’Connell (2020), Toril, Reales, & Ballesteros (2014), Li et al., (2011) for other examples. Aggregate evidence seems to point towards diminished returns after about five hours of training.

---

## [Decision Letter · Decision Letter 1]

18 Nov 2022

PONE-D-22-06647R1A Game-Factors Approach to Cognitive Benefits from Video-Game training: A Meta-AnalysisPLOS ONE

Dear Dr. Basak,

Thank you for submitting your revised manuscript to PLOS ONE. I have sent the manuscript back to the reviewers of the first submission. In general, they were thankful for your serious consideration of their comments and the corresponding improvements undertaken. Yet, the reviewers still have a number of concerns that require further revision. While Reviewer 2 has only one point for clarification, Reviewer 1 made a more severe list of issues. Addressing these issues seem relevant to increase the impact of the paper and to guarantee a better alignment of the proposals made in the current paper, the past literature, and also were the field is heading to. The comments made by Reviewer 1 are manifold, but are very constructive. I encourage you to address the points to the best of your abilities.

We look forward to receiving your revised manuscript.

Kind regards,

Alessandra S. Souza, Ph.D.

Academic Editor

PLOS ONE

Reviewers' comments:

Reviewer's Responses to Questions

**Comments to the Author**

1. If the authors have adequately addressed your comments raised in a previous round of review and you feel that this manuscript is now acceptable for publication, you may indicate that here to bypass the “Comments to the Author” section, enter your conflict of interest statement in the “Confidential to Editor” section, and submit your "Accept" recommendation.

Reviewer #1: (No Response)

Reviewer #2: (No Response)

2. Is the manuscript technically sound, and do the data support the conclusions?

Reviewer #1: Yes

Reviewer #2: Partly

3. Has the statistical analysis been performed appropriately and rigorously? 

Reviewer #1: I Don't Know

Reviewer #2: Yes

4. Have the authors made all data underlying the findings in their manuscript fully available?

Reviewer #1: Yes

Reviewer #2: Yes

5. Is the manuscript presented in an intelligible fashion and written in standard English?

Reviewer #1: Yes

Reviewer #2: Yes

6. Review Comments to the Author

Reviewer #1: First I would like to thank the authors for addressing my former remarks. I found the manuscript much improved and could find any information I looked for. However, I am still a bit confused regarding the Overall Cognition Factor, what does this mean and does the fact that also all other outcomes are then analysed. In my view the Overall Cognition effect sizes needs better motivation or should be left out.

Reviewer #2: I would like to reiterate my strong interest in the approach and results. I sincerely believe the analysis of video game features is a great and promising improvement over the game genres approach, and a timely one as the early conceptualization of video game genres needs to be reconsidered to reflect the diversification of video game genres (and of the whole video game ecosystem). Concerning the revised manuscript, the authors have properly addressed most of my concerns, especially regarding the meta-regression analysis. However, there are a few conceptual (theoretical) as well as methodological issues that remain unresolved and that, once clarified, could improve the paper’s impact.

Main comments

1. Game genre and format

First and foremost, I think the coding of game genres and format could be improved. The dichotomic conceptualisation of genres and formats is not only inconsistent with a growing literature on the evolution of video games reviewed in the introduction (lines 47-48): “a coarse distinction between gaming genres is insufficient to describe the profile of cognitive demands of a given game, as modern video games increasingly include features of multiple genres [7].” , but also, and more problematically, the chosen categories for genres (Action vs Strategy) and formats (Casual vs Serious) do not align with the existing literature and with the evolution of the field over the past decades (e.g., Dale et al., 2020; Dale & Green, 2017). Forcing all games to fit into these 2x2 categories is likely to result in blurred and overlapping categories with high heterogeneity, which could account for the lack of significant effects of these 2 factors. To illustrate my point, I was surprised to see games like “Tetris” or “Wii Fit Segway Circuit” be classified as Action-Casual. In contrast, games most commonly associated with the action genre, which are mostly first person shooter games, were found in the Action-Serious category which also included games like Fifa or super mario or pac-man. Such coding is only likely to add to the confusion the authors initially denounce in their introduction (e.g., note here that “Tetris” also appears in the strategy-casual category).

Second, the discussion relies heavily on non-significant effects, which should not be interpreted as evidence for the absence of a difference. For example, the non-significant effect of game genre (and format) may not be surprising considering how games genres and formats were coded in this meta-analysis – which deviates from the definitions of genre and format used in primary studies and in other meta-analyses too. More importantly, the authors rely on the lack of significant effects to criticize and thus diminish the importance of earlier work conducted more than 1 or 2 decades ago, when only a small number of genres were sufficient to appropriately capture meaningful differences between video games in terms of mechanics or gameplay features (and their associated cognitive demands).

In sum, the analysis of game genre and format needs to be improved. At the very minimum, the authors need to fully address the limitations of their choice of grouping and categories and how it is inconsistent with previous work. Alternatively, given the confusion it is likely to add, this analysis could be dropped., In all cases, the discussion should be significantly altered to (i) not over-interpret non-significant effects and (ii) acknowledge that the 2x2 coding approach taken in this work does not reflect the complexity of the video game ecosystem to date and, for the most part, does not align with the categorization of games used in primary studies.

2. Game features

I particularly liked the idea of analyzing video game features, which is the main strength of this study, although the pattern of inter-correlations calls for a cautious interpretation of the results. This actually raised the question of how were these correlations computed since all variables are coded as binary (0 or 1)? What tests were conducted and what type of correlation coefficients are reported in Table 2?

While these results are an important step forward, they do not necessarily invalidate past research, unlike what the authors suggest. Indeed, many of that past research was conducted at a time when games could still be unanimously classified into a relatively small number of distinct and homogeneous genres.

In this respect, including the variable “study group” in the analysis of game features, as well as in the analysis of genre and format seems an unmotivated choice. More generally, it is unclear how genre, format and group are related to game features. Alos, what was the rationale for including study group only and not genre and format in the analysis of game features? Are these three factors correlated and how do they correlate with game features? Adding all 3 variables to Table 2 and showing how strongly video game genre, video game format and study group correlate with game features would be essential to better understand and interpret the pattern of effects (see point 3).

The discussion of the associations between individual game features and cognitive outcomes suggests that some earlier theories may have been misinterpreted. On multiple occasions, the authors misreport the action games literature as predicting that specific features are responsible for their cognitive effects. My reading of this literature is that it has put the emphasis on identifying the combination of features that distinguishes action games from other games that do not produce similar cognitive benefits. This question could be addressed by computing the similarity between games across a number of critical features such as those coded here. Recent theories actually propose that the effects of action games on cognition arise from the particular combination of features that characterize these games (including those coded here and many others such as the presence of variable rewards, the scaffolding of difficulty and challenge to keep players in their zone of proximal development, etc.). Importantly, the action video game literature does not predict that each and every feature in isolation has an effect, quite on the contrary (Bavelier & Green, 2019; Cardoso-Leite, P. et al., 2020). “Crucially, it has been our experience that each of these three characteristics on its own does not guarantee cognitive impact, at least when it comes to attentional control enhancements. Rather, action video games are unique in that they naturally layer these three game characteristics within the same overarching game play. For example, games that put a premium on just one characteristic such as pacing do not seem to similarly enhance attentional control and other aspects of cognition (e.g., Tetris).” (Green & Bavelier 2019, page 156).

This misreading of the literature is only briefly mentioned in the discussion, lines 581-584 (typo included): “Importantly, of the central predictions of the originating from the action game training literature – that combat-focused games facilitate cognitive transfer to both attention/perception and higher-order cognitive functions [1, 25] – is in agreement with the findings of the current meta-analysis.” This is not only a misinterpretation of the action game training literature that has not claimed such type of feature-specific effect on cognition, but also a misleading terminology to reduce action games to combat-games – e.g., fighting games are not expected to have the same impact as action-shooter games. Moreover, the next sentence further highlights inconsistencies between the interpretation of game features and the fact that the adopted coding of game genres is highly uncommon (see also point 1): Lines 484-588: “However, 50% of the “action” games sampled in this study did not feature combat as a primary gameplay mechanism, whereas 40% of the examined “strategy” games did feature combat as a primary interaction, meaning that the finding that combat games facilitate transfer to both of these cognitive outcomes is not a genre-dependent one.” Taken together, these two statements illustrate a main problem with this paper which needs to be addressed: that the coding of game genres applied in this meta-analysis does not align with that used so far in the field.

3. Study group

The effect of “study group” is interesting, although not unexpected and consistent with prior meta-analyses focusing on video game interventions with active control groups. However, it remains unclear why this effect is indicative of the presence of a selection bias in past research and may thus contribute to publication bias (lines 466-484). The finding of stronger effect sizes in the experimental group is expected and doesn’t mean that the choice of control groups was intentionally biased (which the authors seem to interpret as a form of questionable research practice contributing to publication bias). There are two main problems with this claim.

First, the present meta-analysis focuses on within-subject pre-post intervention effects, which are subject to important confounds (e.g., expectations, test-retest, practice or placebo effects), and are known to cause biases in meta-analytic investigations (e.g., see Cuijpers et al., 2017 for a detailed discussion of the limitations of this approach). I understand that this was necessary as the coding of game features required computing separate effect sizes for the experimental and control games. However, the limitations of this approach should be better acknowledged, especially given the strong interpretation the authors draw from the result.

Second, the use of an active control group is critical for establishing causality and also necessary to rule-out non-specific effects and confounds. Importantly, the choice of control group is guided by the specific research question(s) addressed, which have evolved as the field has matured. To date, most active-control groups involve playing a commercially available video game in order to control for unspecific effects that may be induced by video game play per se (e.g., changes in affect or mood due to playing a video game for example). The choice of control games is thus driven to maximize across the dimensions or features (e.g., combinations of gameplay and mechanics) that are hypothesized to play a key role in driving the cognitive effect of the experimental game, while keeping equated important non-specific features (e.g., the very act of playing, positive feedback within the game, engagement with the training, social stimulation, etc). Despite a few exceptions in which the authors explicitly attempted to test whether the presence or absence of specific game features were critical (e.g., Oei & Patterson, 2015), the studies included in this meta-analysis did not seek to isolate a single video game feature, nor did they argue that a specific feature alone was causally responsible for the particular cognitive effects observed (see also Ben-Sadoun & Alvarez, 2022; Choi et al., 2020 for recent developments in this direction).

In all, the discussion about selection bias reads as if the authors were assigning the wrong intention to the primary studies and thus reflects a misunderstanding of past research. To my knowledge, most studies included here sought to either replicate or clarify the extent of effects by delineating the particular types of games and cognitive domains impacted. To demonstrate the effectiveness of an intervention, primary studies assess whether greater benefits in the experimental group are found compared to the control group (the choice of control group is important to test hypotheses about the mechanisms of cognitive improvement). Tellingly, finding a larger effect in the experimental as compared to the control group does not mean that the control game did not produce any effect (as written line 479-480, but contradicted line 633-635) but instead that the effect of the experimental game was stronger than that of the control game. This whole section needs to be clarified to avoid misleading conclusions.

Additional comments

4. The preregistration is lacking critical information about the analysis plan (meta-analytic models), hypotheses tested and expected results. This makes the preregistration less useful as it is not clear what represents a deviation from the planned protocole and still leaves researchers enough degrees of freedom and flexibility in the analysis and reporting of their results.

5. Genre (action vs strategy) and Format (serious vs casual) are first introduced as 4 types or categories of video games, and then analyzed as two separate and independent (orthogonal) factors. While I understand the rationale of casual games spanning various genres, I don’t understand the dichotomy between casual and serious and what was predicted. Here, only the serious-FPS category seems to correspond to what has been called action video games in past research and contains mostly first and third person shooters as typically used in action games interventions (which would show here as an interaction effect). A consequence of this dichotomous view of game genres is that some games do not fit in their assigned category. For example, the games Tetris, Wii Fit Segway Circuit, Angry Birds, Balance, Centipede, Pacman, marble madness, FIFA 2010, Pinball Hall of Fame, MultiTask, or New Super Mario Bros are labeled as action games despite being reported as non-action (mostly control) games in the primary studies. This is at best confusing and at worse likely to set the field on the wrong path.

6. I also wonder whether genre and format are meant to be orthogonal dimensions or simply different attributes. The classification proposed by Simons only considers 3 categories: action, strategy and casual.While I understand the authors’ argument for casual games that span distinct genres, I don’t think this applies to serious games --a term that has been more commonly used to refer to educational or therapeutic games, in contrast to entertainment video games. And indeed calling commercially available FPS or TPS, serious games is likely to be highly confusing to all readers.

7. When the authors state that the genres have become more “blurry” (line 598), it would be helpful to clarify that this is due to both an increase in the number of genres and subgenres with the classification becoming more granular with more genres (rather than less) and narrower subgenres (e.g. FPS is a subgenre of action), together with greater overlap between genres and subgenres. This state of affairs has led to the emergence of hybrid-genres such as action-role-playing games that mix action mechanics with role playing features.

8. The analysis separating by outcome is interesting but may be underpowered and should thus be reported as exploratory or at least interpreted cautiously.

9. I found surprising that Movement and Perspective produce opposite effects in Table 3, given that they are strongly and positively correlated in Table 2. Could the author comment?

10. I could not find the reference to Valdez 2011. Was this study only a 15 minutes intervention as suggested by the study label in the excel file (Experimental Study, 15 minute RDR-NV group)?

11. Line 590: Did you mean coarse instead of course?

Conclusion

In conclusion, the present results do not support the following conclusions that

(i) game genre is a useless construct; it has at least been valid and instrumental in guiding hypothesis generation and testing,

(ii) the analysis of games features invalidates the theories about the action video game effects,

(iii) the choice of control games reflects an experimenter bias (implying a form of questionable research practice).

As a consequence, I would ask to rephrase and downplay several strong judgmental statements in the discussion:

Lines 524-530: “However, games with passive thresholds or objectives better facilitate transfer to AP outcomes than did games with active opponents, and the AP construct was insensitive to the presence or absence of time pressure. This finding provides evidence that some of the post-hoc justifications given in the past explaining the link between “action” video game play and enhanced attention and perception may be inaccurate – specifically, it does not seem that active opponents or time pressure are necessary elements of gameplay to drive attention & perception outcomes.”

Lines 630-640: “This analysis, in line with previous reviews [2] confirmed the presence of publication bias within the video game training field. Our finding with regards to transfer from experimental groups vs. active control groups using VG interventions may shed light on this problem. Based on our findings that videogames utilized in active control group failed to produce cognitive transfer regardless of their gameplay properties, we can reasonably conclude that some form of bias is suppressing significant transfer in those groups. Addressing this bias is imperative in its own right, and may contribute to the reduction of publication bias in future game training interventions. The authors suggest that future studies examine multiple games of various cognitive profiles with the assumption that differential transfer will be observed between groups (i.e. to differing measures of transfer), rather than the a-priori assumption that one condition will be less effective than another [15, 22, 23, 58].

Lines 499-510: “While games utilizing these gameplay features are common in the field (“first-person-shooter” games utilize a first-person perspective, “real-time-strategy” and many “platformer” and “puzzle” games utilize allocentric movement), neither the first-person perspective nor allocentric movement style have been theorized to have a strong impact on cognitive transfer resulting from VGT interventions. Interestingly, no single game examined in the present meta- analysis featured both an allocentric movement style and a 1st-person perspective, though in theory these results suggest that a game featuring both would be a candidate for effective cognitive intervention.”

Lines 549-552: “As was the case with transfer to the AP construct, transfer to the HC construct was insensitive to the presence or absence of time pressure, which has been invoked as a crucial factor driving cognitive transfer from both “Action” and “Strategy” game interventions [19, 25].“

Lines 576-581: “The findings of the present meta-analysis demonstrate the limited utility of broad genre classifications in understanding the effects of videogame training. Not only did the “action” vs

“strategy” distinction prove ineffective in distinguishing cognitive outcomes of VGT-based cognitive interventions, the gameplay features where were found to differentially impact cognitive outcomes to general cognition, attention/perception, and higher-order cognition did not correspond to defining features of either genre.”

Lines 581-582: Importantly, of the central predictions of the originating from the action game training literature – that combat-focused games facilitate cognitive transfer to both attention/perception and higher-order cognitive functions [1, 25]”

7. PLOS authors have the option to publish the peer review history of their article (what does this mean?). If published, this will include your full peer review and any attached files.

Reviewer #1: No

Reviewer #2: No

---

## [Author Response · Author response to Decision Letter 1]

4 Apr 2023

The reviewers’ comments are in italics. The response are below the comments.

Reviewer #1’s Comment: First I would like to thank the authors for addressing my former remarks. I found the manuscript much improved and could find any information I looked for. However, I am still a bit confused regarding the Overall Cognition Factor, what does this mean and does the fact that also all other outcomes are then analysed. In my view the Overall Cognition effect sizes needs better motivation or should be left out.

a. The authors have revised the manuscript w.r.t. to overall cognition. We believe that in line with the pre-registration, standard reporting of cognitive training on cognition, and past mixed results on videogame interventions effects on cognition, the reporting of overall cognition is important and has a strong value, esp. where the moderator effects are observed. 

Reviewer #2 Comments: I would like to reiterate my strong interest in the approach and results. I sincerely believe the analysis of video game features is a great and promising improvement over the game genres approach, and a timely one as the early conceptualization of video game genres needs to be reconsidered to reflect the diversification of video game genres (and of the whole video game ecosystem). Concerning the revised manuscript, the authors have properly addressed most of my concerns, especially regarding the meta-regression analysis. However, there are a few conceptual (theoretical) as well as methodological issues that remain unresolved and that, once clarified, could improve the paper’s impact.

Main Comments:

1. First and foremost, I think the coding of game genres and format could be improved. The dichotomic conceptualisation of genres and formats is not only inconsistent with a growing literature on the evolution of video games reviewed in the introduction (lines 47-48): “a coarse distinction between gaming genres is insufficient to describe the profile of cognitive demands of a given game, as modern video games increasingly include features of multiple genres [7].” , but also, and more problematically, the chosen categories for genres (Action vs Strategy) and formats (Casual vs Serious) do not align with the existing literature and with the evolution of the field over the past decades (e.g., Dale et al., 2020; Dale & Green, 2017). Forcing all games to fit into these 2x2 categories is likely to result in blurred and overlapping categories with high heterogeneity, which could account for the lack of significant effects of these 2 factors. To illustrate my point, I was surprised to see games like “Tetris” or “Wii Fit Segway Circuit” be classified as Action-Casual. In contrast, games most commonly associated with the action genre, which are mostly first person shooter games, were found in the Action-Serious category which also included games like Fifa or super mario or pac-man. Such coding is only likely to add to the confusion the authors initially denounce in their introduction (e.g., note here that “Tetris” also appears in the strategy-casual category).

a. The authors entirely agree with reviewer 2 that these binarized genre classifications are defined on inherently indistinct categories, and have likely resulted in very heterogeneous games being classified within a given category. We disagree however that these distinctions “do not align with the existing literature”. Our selection of “action” vs “strategy” and “casual” vs “serious/non-casual” was primarily based on Simons et al., 2016’s review of video-game training literature, which as reviewer 2 alluded to identified “Action”, “Strategy”, and “Casual” as the most common labels applied in the literature at the time. While the cited papers by Dale and colleagues rightly assert the limited utility of genre classification, we assert that the vast majority of the literature still utilizes the genre classifications that we have binaries and encoded in this meta-analysis, particularly as the majority of the studies we cite were published before the Dale articles. In short, we agree that these categories do not properly reflect meaningful differences in gameplay profiles, but do accurately reflect how these terms are (mis)applied in the bulk of the literature we are reviewing, and hence serve as a meaningful “control” to the game factors approach we recommend.

b. Re the examples of Tetris and Segway Circuit

i. “Tetris” and “Super Tetris” were both categorized as strategy-casual by our raters (see supplementary table 1 and our shared dataset). The only reference to “Tetris” in relation to “action” games as far as the authors are aware is the mention that “Tetris” is often used as an active control game in action game interventions (alongside “The Sims”). 

ii. In terms of gameplay, Segway Circuit is a time trial racing game, strongly resembling other racing games which have been lumped under the “action” game umbrella such as “Need for Speed”, “Gran Turismo”, and “Mario Kart”, with the major difference that those games may (but don’t necessarily, based on the game mode selected) feature active opponents. In our opinion, it is not surprising that the majority of our raters coded “Segway Circuit” as an “Action” game based on these characteristics.

2. Second, the discussion relies heavily on non-significant effects, which should not be interpreted as evidence for the absence of a difference. For example, the non-significant effect of game genre (and format) may not be surprising considering how games genres and formats were coded in this meta-analysis – which deviates from the definitions of genre and format used in primary studies and in other meta-analyses too. More importantly, the authors rely on the lack of significant effects to criticize and thus diminish the importance of earlier work conducted more than 1 or 2 decades ago, when only a small number of genres were sufficient to appropriately capture meaningful differences between video games in terms of mechanics or gameplay features (and their associated cognitive demands).

a. This is a fair criticism, and our discussion has been revised to more soberly interpret these non-significant findings (as further specified in comments below). 

b. As an aside, we disagree with the assertion that “only a small number of genres were sufficient to appropriately capture meaningful differences between video games... one or two decades ago”. While we go into some depth re. this opinion in other specific responses (#13, among others), fully addressing this is outside the scope of these responses. The primary author (Evan T. Smith) would happily continue this discussion with reviewer 2 after the review process is complete, should they wish to do so.

3. In sum, the analysis of game genre and format needs to be improved. At the very minimum, the authors need to fully address the limitations of their choice of grouping and categories and how it is inconsistent with previous work. Alternatively, given the confusion it is likely to add, this analysis could be dropped., In all cases, the discussion should be significantly altered to (i) not over-interpret non-significant effects and (ii) acknowledge that the 2x2 coding approach taken in this work does not reflect the complexity of the video game ecosystem to date and, for the most part, does not align with the categorization of games used in primary studies.

a. The limitations of our approach have been more strongly emphasized in our discussion, as detailed in later comments. In regards to dropping this analysis, we feel, as elaborated in our response to comment #1, that our coding of genre does reflect how genre has been (mis)applied in the bulk of past literature, and feel it is therefore a valuable inclusion in this manuscript

4. I particularly liked the idea of analyzing video game features, which is the main strength of this study, although the pattern of inter-correlations calls for a cautious interpretation of the results. This actually raised the question of how were these correlations computed since all variables are coded as binary (0 or 1)? What tests were conducted and what type of correlation coefficients are reported in Table 2?

a. Spearman’s correlation coefficient is reported on Table 2 (though in the case of binarized data that is equivalent to Pearson’s or Kendal’s tests). This is admittedly not the most rigorous way of assessing patterns of inter-correlations, but it does convey the fact that these factors are inter-related in a complex way, which as reviewer 2 mentions is important in interpreting our results.

b. As with the coded game factors themselves, a more rigorous mapping of how game factors relate to one another is a next logical step in this line of research, which the authors are keen to pursue in future manuscripts.

5. While these results are an important step forward, they do not necessarily invalidate past research, unlike what the authors suggest. Indeed, many of that past research was conducted at a time when games could still be unanimously classified into a relatively small number of distinct and homogeneous genres.

a. See response to comments 2 and 11.

6. In this respect, including the variable “study group” in the analysis of game features, as well as in the analysis of genre and format seems an unmotivated choice. More generally, it is unclear how genre, format and group are related to game features. Alos, what was the rationale for including study group only and not genre and format in the analysis of game features? Are these three factors correlated and how do they correlate with game features? Adding all 3 variables to Table 2 and showing how strongly video game genre, video game format and study group correlate with game features would be essential to better understand and interpret the pattern of effects (see point 3).

a. Regarding the relationship between genre, format, and game features – indeed we agree that the relationship of specific game features to these commonly-used genre distinctions are indistinct and by-and-large arbitrary, which is why we propose the game factors approach as a more viable alternative. 

b. Regarding the logic of excluding genre and formant form the game factors model – we conceptualize the game factors approach and the genre approach as two different ways of categorizing the same complex phenomenon – that is the wide range of possible gameplay styles present in commercial video games. We’re not interested in seeing how gameplay factors relate to cognition above-and-beyond genre – we’re interested in which of these methods more accurately relate to cognitive outcomes of training. Hence, we ran models featuring each approach separately, and then compared model validity between them (via AIC in this case).

c. Genre/Format have been added to Table 2.

7. The discussion of the associations between individual game features and cognitive outcomes suggests that some earlier theories may have been misinterpreted. On multiple occasions, the authors misreport the action games literature as predicting that specific features are responsible for their cognitive effects. My reading of this literature is that it has put the emphasis on identifying the combination of features that distinguishes action games from other games that do not produce similar cognitive benefits. This question could be addressed by computing the similarity between games across a number of critical features such as those coded here. Recent theories actually propose that the effects of action games on cognition arise from the particular combination of features that characterize these games (including those coded here and many others such as the presence of variable rewards, the scaffolding of difficulty and challenge to keep players in their zone of proximal development, etc.). Importantly, the action video game literature does not predict that each and every feature in isolation has an effect, quite on the contrary (Bavelier & Green, 2019; Cardoso-Leite, P. et al., 2020). “Crucially, it has been our experience that each of these three characteristics on its own does not guarantee cognitive impact, at least when it comes to attentional control enhancements. Rather, action video games are unique in that they naturally layer these three game characteristics within the same overarching game play. For example, games that put a premium on just one characteristic such as pacing do not seem to similarly enhance attentional control and other aspects of cognition (e.g., Tetris).” (Green & Bavelier 2019, page 156).

a. We thank reviewer 2 for this valuable input, and have adjusted our discussion accordingly (see comments below), and specifically invoke the idea that our analysis of specific game features does not account for their conjunctive effects (i.e. lines 509-516, 545, 620-621)

8. This misreading of the literature is only briefly mentioned in the discussion, lines 581-584 (typo included): “Importantly, of the central predictions of the originating from the action game training literature – that combat-focused games facilitate cognitive transfer to both attention/perception and higher-order cognitive functions [1, 25] – is in agreement with the findings of the current meta-analysis.” This is not only a misinterpretation of the action game training literature that has not claimed such type of feature-specific effect on cognition, but also a misleading terminology to reduce action games to combat-games – e.g., fighting games are not expected to have the same impact as action-shooter games. Moreover, the next sentence further highlights inconsistencies between the interpretation of game features and the fact that the adopted coding of game genres is highly uncommon (see also point 1): Lines 484-588: “However, 50% of the “action” games sampled in this study did not feature combat as a primary gameplay mechanism, whereas 40% of the examined “strategy” games did feature combat as a primary interaction, meaning that the finding that combat games facilitate transfer to both of these cognitive outcomes is not a genre-dependent one.” Taken together, these two statements illustrate a main problem with this paper which needs to be addressed: that the coding of game genres applied in this meta-analysis does not align with that used so far in the field.

a. As stated in comment 1 and elsewhere in these responses, we disagree that the coding of game genres differs from what is used I the field, though certainly not all publications conform to this action/strategy or action/nonaction dichotomy. That being said, the point re conflating action games with combat is well-taken. We have modified several sections throughout the manuscript to correct this (i.e. lines 600-604 & those sections mentioned in reviewer comment #18)

9. The effect of “study group” is interesting, although not unexpected and consistent with prior meta-analyses focusing on video game interventions with active control groups. However, it remains unclear why this effect is indicative of the presence of a selection bias in past research and may thus contribute to publication bias (lines 466-484). The finding of stronger effect sizes in the experimental group is expected and doesn’t mean that the choice of control groups was intentionally biased (which the authors seem to interpret as a form of questionable research practice contributing to publication bias). There are two main problems with this claim. First, the present meta-analysis focuses on within-subject pre-post intervention effects, which are subject to important confounds (e.g., expectations, test-retest, practice or placebo effects), and are known to cause biases in meta-analytic investigations (e.g., see Cuijpers et al., 2017 for a detailed discussion of the limitations of this approach). I understand that this was necessary as the coding of game features required computing separate effect sizes for the experimental and control games. However, the limitations of this approach should be better acknowledged, especially given the strong interpretation the authors draw from the result. Second, the use of an active control group is critical for establishing causality and also necessary to rule-out non-specific effects and confounds. Importantly, the choice of control group is guided by the specific research question(s) addressed, which have evolved as the field has matured. To date, most active-control groups involve playing a commercially available video game in order to control for unspecific effects that may be induced by video game play per se (e.g., changes in affect or mood due to playing a video game for example). The choice of control games is thus driven to maximize across the dimensions or features (e.g., combinations of gameplay and mechanics) that are hypothesized to play a key role in driving the cognitive effect of the experimental game, while keeping equated important non-specific features (e.g., the very act of playing, positive feedback within the game, engagement with the training, social stimulation, etc). Despite a few exceptions in which the authors explicitly attempted to test whether the presence or absence of specific game features were critical (e.g., Oei & Patterson, 2015), the studies included in this meta-analysis did not seek to isolate a single video game feature, nor did they argue that a specific feature alone was causally responsible for the particular cognitive effects observed (see also Ben-Sadoun & Alvarez, 2022; Choi et al., 2020 for recent developments in this direction). In all, the discussion about selection bias reads as if the authors were assigning the wrong intention to the primary studies and thus reflects a misunderstanding of past research. To my knowledge, most studies included here sought to either replicate or clarify the extent of effects by delineating the particular types of games and cognitive domains impacted. To demonstrate the effectiveness of an intervention, primary studies assess whether greater benefits in the experimental group are found compared to the control group (the choice of control group is important to test hypotheses about the mechanisms of cognitive improvement). Tellingly, finding a larger effect in the experimental as compared to the control group does not mean that the control game did not produce any effect (as written line 479-480, but contradicted line 633-635) but instead that the effect of the experimental game was stronger than that of the control game. This whole section needs to be clarified to avoid misleading conclusions.

a. We thank reviewer 2 for this thorough and fair critique of our conclusions re. the experimental/control group findings. As with most of our discussion, we have reworked the sections in question to more accurately reflect the ambiguity in these findings that Reviewer 2 has highlighted here (see lines 482-495)

10. The preregistration is lacking critical information about the analysis plan (meta-analytic models), hypotheses tested and expected results. This makes the preregistration less useful as it is not clear what represents a deviation from the planned protocol and still leaves researchers enough degrees of freedom and flexibility in the analysis and reporting of their results.

a. We thank the reviewer about this insight and opinion. At the time of pre-registration, the focus was mainly on the main aims regarding game factors meta and the inclusion/exclusion criterion. We therefore believe that the pre-registration is still important than non-registered meta-analysis in terms of the main goals of the project and the specifics of I/E criteria in the broad field of videogame. 

11. Genre (action vs strategy) and Format (serious vs casual) are first introduced as 4 types or categories of video games, and then analyzed as two separate and independent (orthogonal) factors. While I understand the rationale of casual games spanning various genres, I don’t understand the dichotomy between casual and serious and what was predicted. Here, only the serious-FPS category seems to correspond to what has been called action video games in past research and contains mostly first and third person shooters as typically used in action games interventions (which would show here as an interaction effect). A consequence of this dichotomous view of game genres is that some games do not fit in their assigned category. For example, the games Tetris, Wii Fit Segway Circuit, Angry Birds, Balance, Centipede, Pacman, marble madness, FIFA 2010, Pinball Hall of Fame, MultiTask, or New Super Mario Bros are labeled as action games despite being reported as non-action (mostly control) games in the primary studies. This is at best confusing and at worse likely to set the field on the wrong path.

a. The format variable is fairly rigidly defined via the definition of “casual” games commonly used in the literature, that being games designed to be played for 30 minutes or less in a single session (and “serous”/”long-form” therefore being designed to be played for longer than 30 minutes in a session). We agree, as mentioned in previous comments, that binarizing the action/strategy distinction does not adequately categorize all games, but as discussed in our response to comment #1 we feel this forces categorization accurately reflects the common misapplication of genre terms in the literature. Accurately categorizing the genre of a given game would require a much more granular approach than has been applied in the field to date, and in the end many of those distinctions would be arbitrary (do first-person shooter game and a first-person action game that does not involve projectile combat meaningfully differ in terms of cognitive demands evoked?). The authors assert that a more granular categorization of genre (i.e. “action/strategy/puzzle”) does not address the issues that reviewer 2 raises, whereas a game factors approach does – indeed, that is the thesis of this paper.

12. I also wonder whether genre and format are meant to be orthogonal dimensions or simply different attributes. The classification proposed by Simons only considers 3 categories: action, strategy and casual. While I understand the authors’ argument for casual games that span distinct genres, I don’t think this applies to serious games --a term that has been more commonly used to refer to educational or therapeutic games, in contrast to entertainment video games. And indeed calling commercially available FPS or TPS, serious games is likely to be highly confusing to all readers.

a. We had intended the term “Serious” to refer to the inverse of “Casual”, i.e. games designed to be played for sessions longer than 30 minutes. We have adjusted this term to “long-form” throughout the manuscript, to avoid confusion with the term “serious” game as it has been used to refer to educational/therapeutic games.

13. When the authors state that the genres have become more “blurry” (line 598), it would be helpful to clarify that this is due to both an increase in the number of genres and subgenres with the classification becoming more granular with more genres (rather than less) and narrower subgenres (e.g. FPS is a subgenre of action), together with greater overlap between genres and subgenres. This state of affairs has led to the emergence of hybrid-genres such as action-role-playing games that mix action mechanics with role playing features.

a. We don’t entirely agree that the in lack of distinction of game genres is resultant of “both an increase in the number of genres … and narrower subgenres (e.g. FPS is a subgenre of action), together with greater overlap between genres and subgenres.” Genres have always been a post-hoc categorization of a rich and complex gameplay space, from the earliest days of commercial gaming as a hobby. Stating that “genres are becoming more indistinct” miscategorized the issues – gaming enthusiast, marketers, etc. are continuously developing increasingly specific genre terminology to describe specific points in “gameplay space”, and admittedly this is driven by continual permutation of existing and development of new gameplay tropes in the developer space, but genres have never been accurate summaries of gameplay, (in this author’s opinion)

b. A full exploration of this topic is, as mentioned, outside the scope of this paper (and indeed, the field of cognitive science) – considering this, we have elected to keep our summary of this issue brief, but have adjusted the wording to better convey the nuances of our argument beyond “blurry” (see lines 613-616).

14. The analysis separating by outcome is interesting but may be underpowered and should thus be reported as exploratory or at least interpreted cautiously.

a. We acknowledge that our analysis of separate outcomes is lacking in power particularly with regards to memory and psychosocial outcomes on lines 570-580. We contend that the sub-analyses pertaining to Attention/Perception and Higher-order Cognition, with ks of 90 and 54 respectively, are sufficiently powered, particularly with respect to other meta-analyses.

15. I found surprising that Movement and Perspective produce opposite effects in Table 3, given that they are strongly and positively correlated in Table 2. Could the author comment?

a. Correlation between the two predictor variables does not mean that they need to have the same main effect on the dependent variable, especially after properly controlling for other influences. As each fixed effect in the LME approach is corrected for all other fixed effects, we can state that, for example, the impact of perspective on cognitive transfer is negative, (i.e. favoring 1st-person perspective) all other things being equal, including movement. We would expect, based on these results, that both egocentric games in a 1st-person perspective would produce greater cognitive transfer than egocentric games in a 3rd-person perspective. The correlation simply states that, in addition to this, egocentric games are more likely to be 1st-person whereas allocentric games are more likely to be 3rd-person.

16. I could not find the reference to Valdez 2011. Was this study only a 15 minutes intervention as suggested by the study label in the excel file (Experimental Study, 15 minute RDR-NV group)?

a. The full citations for studies included in the meta-analysis but not directly referenced in the body of the main manuscript can be found in our supplementary references, which for the Valdez study in question is “Valadez, J. J., & Ferguson, C. J. (2012). Just a game after all: Violent video game exposure and time spent playing effects on hostile feelings, depression, and visuospatial cognition. Computers in Human Behavior, 28(2), 608–616. https://doi.org/10.1016/j.chb.2011.11.006”. We note that the year of this study was incorrectly listed as “2011” in our public access data (this has been corrected), but is correctly cited to the year 2012 in the supplementary materials.

b. To answer reviewer 2’s initial question, Valdez included both a 15-minute and a 45-minute intervention group. Indeed this is the shortest intervention included in this meta-analysis.

17. Line 590: Did you mean coarse instead of course?

a. Indeed we did – we thank reviewer 2 for the correction.

18. In conclusion, the present results do not support the following conclusions that(i) game genre is a useless construct; it has at least been valid and instrumental in guiding hypothesis generation and testing, (ii) the analysis of games features invalidates the theories about the action video game effects, (iii) the choice of control games reflects an experimenter bias (implying a form of questionable research practice). As a consequence, I would ask to rephrase and downplay several strong judgmental statements in the discussion:

i. We thank reviewer 2 for their thorough critique of our discussion, and recommendations for improvement. All of the sections listed below have been adjusted in line with reviewer twos feedback – line numbers after revision (as well as specific comments as necessary) are included below.

b. Lines 524-530: “However, games with passive thresholds or objectives better facilitate transfer to AP outcomes than did games with active opponents, and the AP construct was insensitive to the presence or absence of time pressure. This finding provides evidence that some of the post-hoc justifications given in the past explaining the link between “action” video game play and enhanced attention and perception may be inaccurate – specifically, it does not seem that active opponents or time pressure are necessary elements of gameplay to drive attention & perception outcomes.”

i. This section has been reworked in line with reviewer 2’s suggestions (lines 538-545)

c. Lines 630-640: “This analysis, in line with previous reviews [2] confirmed the presence of publication bias within the video game training field. Our finding with regards to transfer from experimental groups vs. active control groups using VG interventions may shed light on this problem. Based on our findings that videogames utilized in active control group failed to produce cognitive transfer regardless of their gameplay properties, we can reasonably conclude that some form of bias is suppressing significant transfer in those groups. Addressing this bias is imperative in its own right, and may contribute to the reduction of publication bias in future game training interventions. The authors suggest that future studies examine multiple games of various cognitive profiles with the assumption that differential transfer will be observed between groups (i.e. to differing measures of transfer), rather than the a-priori assumption that one condition will be less effective than another [15, 22, 23, 58].

i. This paragraph has been removed. Discussion of selection bias is now relegated to lines 468-488 and presents, we believe, a more balanced perspective on the findings that intervention games consistently produced greater transfer than control games in this analysis.

d. Lines 499-510: “While games utilizing these gameplay features are common in the field (“first-person-shooter” games utilize a first-person perspective, “real-time-strategy” and many “platformer” and “puzzle” games utilize allocentric movement), neither the first-person perspective nor allocentric movement style have been theorized to have a strong impact on cognitive transfer resulting from VGT interventions. Interestingly, no single game examined in the present meta- analysis featured both an allocentric movement style and a 1st-person perspective, though in theory these results suggest that a game featuring both would be a candidate for effective cognitive intervention.”

i. This section has been reworked in accordance with reviewer 2’s comments (lines 505-524)

e. Lines 549-552: “As was the case with transfer to the AP construct, transfer to the HC construct was insensitive to the presence or absence of time pressure, which has been invoked as a crucial factor driving cognitive transfer from both “Action” and “Strategy” game interventions [19, 25].”

i. We contend that this statement re. the insensitivity of the HC construct to time pressure as well as the invocation of time pressure as a driver of cognitive change in past work is accurate. This has been reworded slightly so as to be less strongly assertive.

f. Lines 576-581: “The findings of the present meta-analysis demonstrate the limited utility of broad genre classifications in understanding the effects of videogame training. Not only did the “action” vs “strategy” distinction prove ineffective in distinguishing cognitive outcomes of VGT-based cognitive interventions, the gameplay features where were found to differentially impact cognitive outcomes to general cognition, attention/perception, and higher-order cognition did not correspond to defining features of either genre.”

i. The “Relevance of Existing Genre Definitions…” (beginning on line 590) of which this quoted section was the opening to has been majorly re-worked in line with Reviewer 2’s suggestion.

g. Lines 581-582: Importantly, of the central predictions of the originating from the action game training literature – that combat-focused games facilitate cognitive transfer to both attention/perception and higher-order cognitive functions [1, 25]”

i. We feel this statement is broadly accurate - we do in fact see a positive impact on transfer of combat-based games, which aligns with previous literature. This statement is not based on any null effect (which reviewer 2 rightly criticized with regards to other strong statements in our discussion). While reviewer 2 rightly criticized us for conflating combat to the “action” genre in an earlier comment, we feel that this statement – specifically that the combat-focused nature of many action games may be a facilitator of transfer from training with those games – does reflect a fairly common opinion present in the literature.

---

## [Decision Letter · Decision Letter 2]

5 May 2023

A Game-Factors Approach to Cognitive Benefits from Video-Game training: A Meta-Analysis

PONE-D-22-06647R2

Dear Dr. Basak,

We’re pleased to inform you that your manuscript has been judged scientifically suitable for publication and will be formally accepted for publication once it meets all outstanding technical requirements.

Kind regards,

Alessandra S. Souza, Ph.D.

Academic Editor

PLOS ONE

Additional Editor Comments (optional):

Reviewers' comments:

Reviewer's Responses to Questions

**Comments to the Author**

1. If the authors have adequately addressed your comments raised in a previous round of review and you feel that this manuscript is now acceptable for publication, you may indicate that here to bypass the “Comments to the Author” section, enter your conflict of interest statement in the “Confidential to Editor” section, and submit your "Accept" recommendation.

Reviewer #2: All comments have been addressed

2. Is the manuscript technically sound, and do the data support the conclusions?

Reviewer #2: Yes

3. Has the statistical analysis been performed appropriately and rigorously? 

Reviewer #2: Yes

4. Have the authors made all data underlying the findings in their manuscript fully available?

Reviewer #2: No

5. Is the manuscript presented in an intelligible fashion and written in standard English?

Reviewer #2: Yes

6. Review Comments to the Author

Reviewer #2: The reviewers have addressed all my comments and suggestions and have modified their manuscript accordingly. There are still some points of disagreements regarding the conceptualisation and discussion of game genres in the field. However, I believe that disagreements and debates can be beneficial to science, especially when they are respectful and constructive.

7. PLOS authors have the option to publish the peer review history of their article (what does this mean?). If published, this will include your full peer review and any attached files.

Reviewer #2: No

---

## [Editor Report · Acceptance letter]

22 Jun 2023

PONE-D-22-06647R2 

A game-factors approach to cognitive benefits from video-game training: A meta-analysis 

Dear Dr. Basak:

I'm pleased to inform you that your manuscript has been deemed suitable for publication in PLOS ONE. Congratulations! Your manuscript is now with our production department. 

Kind regards, 

on behalf of

Dr. Alessandra S. Souza 

Academic Editor

PLOS ONE